# Bias in modeled Greenland ice sheet melt revealed by ASCAT

Anna Puggaard[1,2], Nicolaj Hansen[2], Ruth Mottram[2], Thomas Nagler[3], Stefan Scheiblauer[3], Sebastian B. Simonsen[1], Louise S. Sørensen[1], Jan Wuite[3], and Anne M. Solgaard[4]

[1]Geodesy and Earth Observation, DTU-Space, Technical University of Denmark, Lyngby, Denmark
[2]National Centre for Climate Research, Danish Meteorological Institute (DMI), Copenhagen, Denmark
[3]Environmental Earth Observation Information Technology (ENVEO) IT GmBH, Innsbruck, Austria
[4]Glaciology and Climate, Geological Survey of Denmark and Greenland (GEUS), Copenhagen, Denmark

**Correspondence:** Anna Puggaard (annpu@space.dtu.dk)

**Abstract.** The runoff of surface melt is the primary driver of mass loss over the Greenland Ice Sheet. An accurate representation of surface melt is crucial for understanding the surface mass balance and, ultimately, the ice sheet's total contribution to sea level rise. Regional climate models (RCMs) model ice-sheet-wide melt volume but exhibit large variability in estimates among models, requiring validation with observed melt. Here, we explore a novel processing of data from the Advanced SCATterome-

ter (ASCAT) instrument onboard the EUMETSAT Metop satellites, which provides estimates of the spatiotemporal variability of melt extent over the Greenland Ice Sheet between 2007 and 2020. We apply the ASCAT surface melt extent maps to pinpoint differences in the melt products from three distinct RCMs, where one is forced at the boundary with two different reanalyses. Using automatic weather station (AWS) air temperature observations, we assess how well RCM-modeled melt volume aligns with in situ temperatures. With this assessment, we establish a threshold for the RCMs to identify how much meltwater is in the

models before it is observed at the AWS and ultimately infer the melt extent in the RCMs. We show that applying thresholds, informed by in situ measurements, reduces the differences between ASCAT and RCMs and minimizes the discrepancies between different RCMs. Differences between modeled melt extent and melt extent observed by ASCAT are used to pinpoint (i) biases in the RCMs, which include variability in their albedo schemes, snowfall, turbulent heat fluxes, and temperature as well as differences in radiation schemes, and (ii) limitations of the liquid water detection by ASCAT, including misclassification in

the ablation zone as well as a temporal melt onset bias. Overall we find the RCMs tend to have a later melt onset than ASCAT and an earlier end of melt season with a similar but slightly smaller melt area than identified in ASCAT. Biases, however, vary spatially between models and with compensating errors in different regions, suggesting that one RCM can sometimes represent the present-day surface across the entire ice sheet more effectively than the ensemble mean.

# 1 Introduction

The Greenland Ice Sheet significantly contributes to the rise in global sea levels, contributing approximately 15 % since the early 1990s (WCRP Global Sea Level Budget Group, 2018). Between 1992 and 2020, satellite observations have shown that the Greenland Ice Sheet has lost $4892 \pm 457$ Gt of ice or $13.6 \pm 1.3$ mm sea level equivalent (Otosaka et al., 2023) with half of the mass loss attributed to a decrease in the surface mass balance (SMB, van den Broeke et al. (2016)). However, the rate of mass loss has exhibited considerable annual variability in recent years, ranging from $86 \pm 75$ Gt in 2017 to $444 \pm 93$ Gt in 2019, with the latter being driven by exceptional surface melting during the summer (Tedesco and Fettweis, 2020). Modeling studies have shown that surface melt on the Greenland Ice Sheet has generally doubled since the 1990s due to a rise in temperature (Tedesco and Fettweis, 2020; van den Broeke et al., 2016). Meanwhile, snow accumulation has remained nearly constant (van den Broeke et al., 2016). In the warm summer months, the surface temperature rises above the melting point of ice, and surface melt occurs. Depending on snow and firn characteristics, meltwater generated at the surface can either collect at the surface and form supraglacial meltwater lakes (Koenig et al., 2015), run off as surface meltwater (Smith et al., 2015) or percolate into the snowpack, where it either refreezes (Forster et al., 2014; Harper et al., 2012) or runs off englacially (Chandler et al., 2013).

At present, regional climate models (RCMs) provide the most comprehensive approach for obtaining ice-sheet-wide estimates of meltwater volumes and runoff, with simulations showing the best agreement with observations (Fettweis et al., 2020). However, these models are influenced by the chosen modeling approach, and substantial disparities persist among models (Rae et al., 2012; Vernon et al., 2013; Fettweis et al., 2020; Glaude et al., 2024). In particular, recent studies suggest that small differences between models at the present day, representing differences in parametrizations, have large effects on projections of melt, runoff and surface mass balance when run into the future, giving greater uncertainty on sea level rise estimates than desirable for climate adaptation purposes (Goelzer et al., 2020). Thus, it is crucial to develop methods to evaluate melt estimates from RCMs against observations of melt to understand these discrepancies and ultimately evaluate which RCMs most realistically model melt (Langen et al., 2017). While melt intensity can be derived from in-situ observations at automatic weather stations (AWS), the sparse distribution of these stations across the ice sheet limits the evaluation of melt estimates beyond local scales (Fausto et al., 2018). On the other hand, satellite remote sensing can observe the presence of meltwater at the upper part of the firn pack, while the melt intensity has not yet been measured successfully from remote sensing. Recent approaches, such as those by Dethinne et al. (2023) and Picard et al. (2022), have assimilated remote sensing datasets into more detailed modeling of surface melt processes to estimate meltwater volume. Although including more observational data generally improves the representation of current surface conditions in the RCMs (Langen et al., 2017; Dethinne et al., 2023), there remains a strong need for independent observational dataset to assess model outputs. Remote sensing satellites provide information on the Greenland Ice Sheet surface melt by observations from the visible to the microwave spectrum, where some of the widely used sensors are Advanced SCATterometer (ASCAT), Sentinel-1, Moderate Resolution Imaging Spectroradiometer (MODIS), and Special Sensor Microwave Imager/Sounder (SSMIS) (Husman et al., 2023). Microwave sensors have the advantage of offering measurements independently of cloud cover, weather conditions, and polar darkness. Over the ice sheets, the backscattering of

the microwave signals from snow and ice depends on roughness geometry and electrical properties, which in turn depends on the physical characteristics of the snow and ice (Wismann, 2000; Long, 2017). During the winter, the backscatter signal can exhibit a gradual decrease due to snow accumulation attenuating the volume scattering in the snowpack. As the temperature increases, meltwater at the surface is introduced, and the backscatter signal experiences a substantial drop. This sensitivity to meltwater has enabled several studies to estimate melt over both ice sheets using passive and active microwave measurement with a threshold method to detect the onset of melt and its extent (Long and Drinkwater, 1994; Wismann, 2000; Ashcraft and Long, 2006; Fettweis et al., 2011; Colosio et al., 2021; Husman et al., 2023). The magnitude of the decrease in backscatter varies due to factors such as the snow water content and the specific properties of the snowpack, such as grain size and the presence of ice layers and lenses, which influence the dielectric properties and roughness geometries (Wismann, 2000; Long, 2017). Refrozen meltwater from the previous melt season can percolate into the firn, leading to the formation of subsurface features such as firn aquifers and ice lenses, which can potentially amplify the backscatter signal prior to the current melting season (Brangers et al., 2020). Further, meltwater in the subsurface can still be detected after refreezing of the surface layer as the low-frequency signals can still penetrate into the refrozen surface layer. Using a threshold method proposed by Ashcraft and Long (2006), Husman et al. (2023) showed that the C-band (4–8 GHz) active microwave sensors detected more melt days than K-band (18-27 GHz) passive microwave sensors in areas with meltwater in the subsurface in Antarctica due to differences in penetration depth. Thus, for a correct identification of surface melt using microwave satellite observations, it is important to account for penetration depth, changes in dielectric properties, and roughness geometries of the snowpack. The formation of subsurface ice features and subsurface penetration can lead to significant misclassification of surface melt if a simple threshold method is used (Ashcraft and Long, 2006; Long, 2017). Instead of using a simple threshold method, ASCAT surface melt extent maps utilize an algorithm that incorporates the temporal behavior of the backscattered signal. With this method, the ASCAT surface melt extent maps can not only detect the presence of liquid water on the surface but also distinguish between melting and subsequent refreezing of the surface meltwater. This makes ASCAT surface melt extent maps a unique product as they allow for a more fair comparison between the observed liquid meltwater extent and the surface melt extent simulated by RCMs. Furthermore, by applying an annual recalibration of the winter signal, the product accounts for the formation of subsurface features from the previous melt season (Nagler et al., 2024). Again, this ensures a better classification of melt signal compared to previous melt extent products from both active and passive microwave measurements over the Greenland Ice Sheet, such as Abdalati and Steffen (1995); Wismann (2000); Nghiem et al. (2001); Tedesco (2007); Fettweis et al. (2011); Colosio et al. (2021).

By using ASCAT surface melt extent maps, we aim to establish a framework for evaluating the performance of RCMs in simulating the temporal variability of present-day melt extent. As RCMs are often calibrated with respect to basin-wide surface mass balance, incorporating an independent satellite dataset like ASCAT surface melt extent maps enables a more comprehensive assessment of model performance. By including HIRHAM5, RACMO2.3p2, and MARv3.12, we assess how well each model captures surface melt patterns, focusing not on internal model parametrizations, e.g. albedo and near-surface temperature, but on the representation of melt extent. Here, we employ a liquid water detection algorithm that integrates the temporal behavior of the backscatter signal by classifying the first- and second-time derivatives. Rather than solely identifying

the period of meltwater presence, the method distinguishes between the surface melting and subsequent refreezing of the surface meltwater. However, this method does not estimate the meltwater volume, meaning we can only use the satellite-observed melt to asses the RCMs' ability to represent the extent of surface melt realistically. Hence, we compare the melt extent observed by ASCAT to the modeled melt extent by RCMs. To ensure that the RCM-modeled melt aligns with the in situ observations, we compare modeled melt volume to observed 2m temperatures at automatic weather stations (AWS) stations to establish the melting threshold (mm of water equivalent per day, mm w.e. day$^{-1}$) to identify how much meltwater is in the models before it is observed at the AWS. Once the melt extent is inferred from the RCMs, it's possible to compare it with the melt extent observed by ASCAT to identify biases within the RCMs and limitations in the ASCAT melt observations.

## 2  Data

### 2.1  PROMICE AWS

The Programme for Monitoring of the Greenland Ice Sheet & Greenland Climate Network, PROMICE GC-net AWS, offers hourly and daily meteorological and glaciological in situ measurements for 54 weather stations on the Greenland Ice Sheet, tundra, and peripheral glaciers (Fausto et al., 2021; How et al., 2022). Here, we include the 34 stations on the Greenland Ice Sheet and have measurements of air temperature between 2007 and 2020. PROMICE GC-net only includes active weather stations, but the historical GC-net data includes several discontinued stations. Here data from the Summit, GITS, and Pertermann ELA stations are included, as they overlap in time with ASCAT data (Steffen et al., 2022; Vandecrux et al., 2023). See Fig. 1 for the location of AWS stations.

### 2.2  Regional climate models

The melt volume from RCMs is derived by closing the surface energy budget. When the skin temperature exceeds 0 °C, additional energy contributes towards melting, resetting the skin temperature to 0 °C (Langen et al., 2015; Noël et al., 2018). However, different model setups such as horizontal and vertical resolutions, and choices of parameters like surface albedo and subsurface schemes impact the surface energy balance simulated within these models and thereby result in different melt volumes. We compare the melt extent observed by ASCAT with the modeled melt extent from three RCMs: HIRHAM5, RACMO2.3p2, and MARv3.12, see Fig. 2 and Fig. 3. A detailed description of model differences is provided in the following.

### HIRHAM5

HIRHAM5 (Lucas-Picher et al., 2012) utilizes a rotated polar grid at $0.05° \times 0.05°$ horizontal resolution, which for the Greenland Ice Sheet corresponds to approximately 5.5 km. HIRHAM5 is developed from the dynamics of the numerical weather forecast model High-Resolution Limited Area Model (HIRLAM, Undén et al. (2002)) combined with the physics from the ECHAM5 general circulation model (Roeckner et al., 2003) to ensure accurate simulation of the surface energy balance. HIRHAM5 is forced on the lateral boundary with 6-hourly global reanalysis temperature, relative humidity, wind

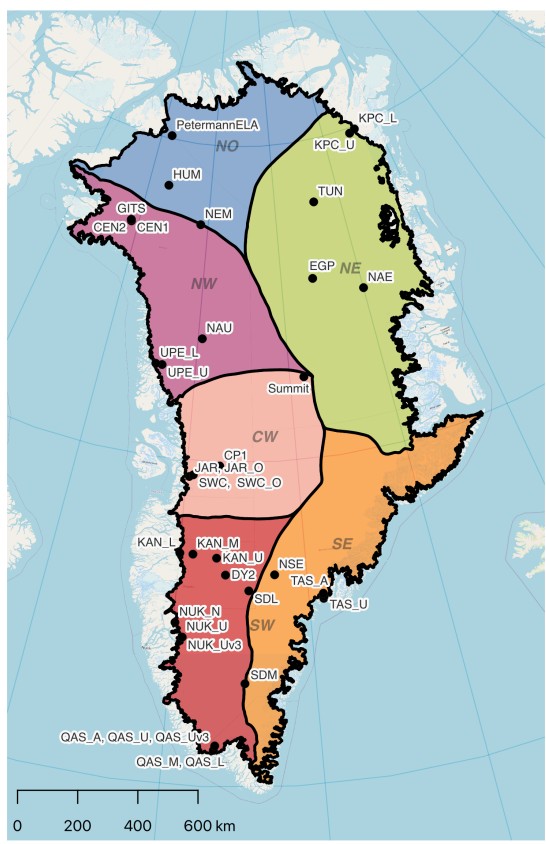

**Figure 1.** Location of AWS stations included in the study. Only AWS stations on the ice sheet (ASCAT liquid water detection domain) are included in this study. Further, included are the Rignot and Mouginot (2012) drainage basins that are utilized in the subsequent evaluation of RCMs modeled melt extent against ASCAT observed melt extent.

vectors, and pressure fields. Further, daily sea ice concentration and sea surface temperature fields are also used to force the model (Langen et al., 2017). Here, we include HIRHAM5 run with both ERA-interim for the period 1979 - 2019, (Dee et al., 2011) and ERA5 for the period 1960 - 2020, (Hersbach et al., 2020). HIRHAM5 forced with ERA-interim is referred to as HIRHAM5-ERAI, and HIRHAM5 forced with ERA5 as HIRHAM5-ERA5. Further, the surface albedo in HIRHAM5-ERAI is derived from MODIS gridded surface albedo (Box et al., 2012), described in (Langen et al., 2017), while the surface albedo in HIRHAM5-ERA5 is computed internally as a linear function of temperature, described in (Langen et al., 2015). The outputs from both HIRHAM5 runs are used to force the offline subsurface model described in Langen et al. (2017). The offline subsurface model includes a multilayer surface snow and mass balance scheme that simulates melt percolation, retention and refreezing with a vertical resolution down to 60 m w.e. (Langen et al., 2017).

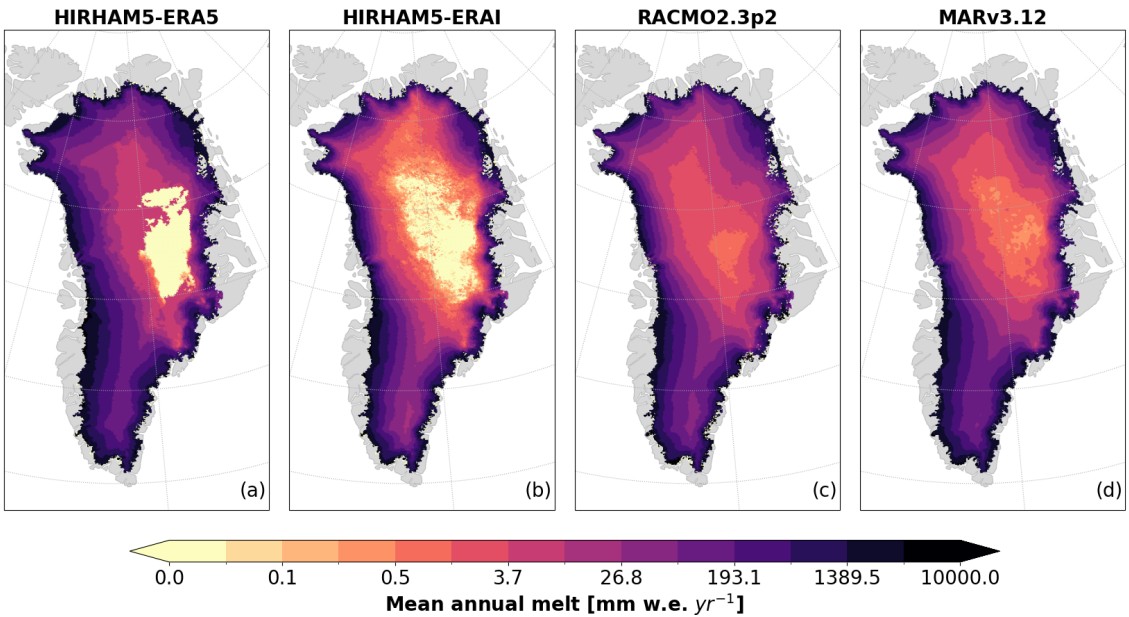

**Figure 2.** Modeled mean annual melt by different RCMs inside the ASCAT data mask (2007-2020).

### RACMO2.3p2

The polar Regional Atmospheric Climate Model (RACMO2.3p2) is run on approximately 5.5 km ($0.05° \times 0.05°$) horizontal resolution (Noël et al., 2018, 2019). RACMO2.3p2 also integrates the numerical weather forecasting dynamics from HIRLAM (Undén et al., 2002) with the physics from the European Centre for Medium-Range Weather Forecasts–Integrated Forecast System cycle CY33r1 (ECMWF, 2009) to accurately simulate the surface energy balance. On the lateral boundary, RACMO2.3p2 uses the same 6-hourly and daily ERA5 fields as HIRHAM5-ERA5 (Hersbach et al., 2020). The snow albedo is derived from snow grain size, cloud optical thickness, solar zenith angle, and impurity concentration in snow. However, to account for temporal variability in the ablation zone, gridded MODIS observations of bare ice (Box et al., 2012) are included in these areas (Van Angelen et al., 2012). The model includes a multilayer snow module for simulating surface melt partitioning into percolation, retention, refreezing, and ultimately runoff (Ettema et al., 2010). The 5.5 km product is statistically downscaled to a horizontal resolution of 1 km to represent the steep SMB gradients found over narrow glaciers and confined ablation zones at the rugged margins of the ice sheet (Noël et al., 2016).

### MARv3.12

The *Modèle Atmosphérique Régional*, MARv3.12, is run at a 10 km horizontal grid resolution (Fettweis et al., 2013, 2017; Tedesco and Fettweis, 2020). MARv3.12 combines atmospheric modeling with a Soil Ice Snow Vegetation Atmosphere Transfer Scheme (SISVATS) (Gallée and Schayes, 1994) to simulate the surface energy balance and mass balance processes over

the ice sheet. The MARv3.12 is forced at the lateral boundaries with ERA5 (Hersbach et al., 2020) at 6-hourly temporal res-
olution. Output values are averaged to obtain daily values (Tedesco and Fettweis, 2020). MARv3.12 includes the snow model
Crocus (Brun et al., 1992), which simulates a number of layers of snow, ice, or firn of variable thickness and energy- and
mass-transports between each layer. The snow model also provides snow grain properties, which are used in combination with
density, age, and type to simulate snow albedo (Brun et al., 1992; Fettweis et al., 2017; Antwerpen et al., 2022), MARv3.12
also have an albedo range for bare ice between 0.4 and 0.55 depending on the cleanliness of the ice (Fettweis et al., 2017).
While both RACMO2.3p2 and HIRHAM5-ERA5 incorporate MODIS observations into the albedo computation, the surface
albedo in MARv3.12 is only based on the internally computed broadband albedo (Brun et al., 1992).

## 2.3 ASCAT surface melt extent maps

The ASCAT instrument onboard the EUMETSAT MetOp polar-orbiting satellites has provided measurements since 2007.
ASCAT is a Real Aperture Radar (RAR) instrument and measures radar backscatter in vertical polarization at a range of in-
cidence angles from 25°to 65°at a frequency of 5.255 GHz (C-band) (Figa-Saldaña et al., 2002; Long, 2017). With a spatial
resolution of 25 km, ASCAT had limited land applications, but with the introduction of the scatterometer image reconstruction
(SIR) algorithm, the utility of ASCAT data has significantly improved (Lindsley and Long, 2010). The SIR algorithm takes
advantage of image reconstruction techniques and the spatial overlap of measurements over a region to enhance the effective
spatial resolution over a time interval. SIR depends on the number of measurements, where increasing the number of mea-
surements leads to a reduction of noise and an improvement of the spatial resolution. However, the radar characteristics must
remain constant between passes during the interval time. We refer to Long et al. (1993) and Long and Drinkwater (1994) for
a more detailed description of the SIR algorithm. For Greenland, a 4-day time interval is appropriate as it ensures sufficiently
dense sampling while justifying the assumption of constant radar characteristics (Lindsley and Long, 2010). With the 4-day
time interval, diurnal variations in backscatter signal over ice, such as melt during the day and refreezing during the night, are
averaged out, which introduces additional uncertainty. Additionally, the resolution-enhanced ASCAT product may not capture
short melt events in the spring and intense precipitation events, as these signals are averaged. Furthermore, potential azimuth
angle dependencies are not considered in the construction of the SIR products (Long and Drinkwater, 1994; Lindsley and Long,
2010).

The ASCAT SIR product is available from 2007 to 2020 and is used to identify four different melt stages by applying a
hierarchical decision tree using dynamic thresholds based on the previous winter reference month (JFM) and the first- and
second-time derivatives of the backscattered signal, $\sigma^0$ (Nagler et al., 2024). During the winter months, the backscatter signal
is relatively stable, and only minor variations may occur due to changes in snowpack properties (label Stage-1 (ST-1), no
melt). To account for the possibility of remnant changes in the snowpack from the previous melt season, the winter signal is
recalibrated annually pixel-by-pixel. As the snow surface starts to melt in spring, rising above liquid water content of $\sim 1\%$
volume (Mätzler, 1987), the backscatter signal decreases significantly. This rapid drop in the backscatter coefficient is utilized
to identify the onset and occurrence of surface melt (label ST-2A, surface melt). At intense melt events, the backscatter signal
becomes fully saturated, which means that an increase in the melt intensity does not lead to further lowering of the backscatter

signal (label ST-2B, wet snow layer). As the meltwater starts to refreeze, the backscatter signal will gradually increase again, returning to stable winter conditions once the meltwater down to the radar penetration depth is refrozen (label ST-3, increase in the refrozen layer). For a more detailed description of the melt stage classification, we refer to (Nagler et al., 2024).

As an example, Fig. 4a shows the backscatter measurements, and the liquid water classification is shown for the KAN_U AWS. The winter dry signal changes after at least one melt season, highlighting the melt classification method's ability to correctly detect liquid water despite a difference in the winter signal before and after the melt season. When comparing the liquid water extent observed by ASCAT and the melt extent model by the RCMs, we excluded label ST-3 (increase in the refrozen layer) since we are only interested in comparing surface melt. Figure 4b illustrates a snapshot of the detection and classification from 07-08-2017. Refreezing or no melting is observed in pixels close to the ice sheet margin when melt occurs at higher elevations on the ice sheet. Further, when looking at the mean annual number of melt days, the lower ablation zone exhibits fewer melt days than at higher elevations. This seemingly incorrect classification of liquid water near the ice margin is an effect of changes in the surface roughness associated with melt. Changes in surface roughness can counteract the decrease in backscatter associated with melt, effectively concealing liquid water in the backscatter signal. Therefore, when comparing ASCAT and RCMs, we mask out these areas by applying the maximum elevation of the snowline between 2007-2020 (Box et al., 2012), where pixels with an elevation below are excluded in the comparison (Fig. 4d).

## 3 Methods

Modeled surface melt in RCMs is subject to large variability among models as seen in Fig. 2 and discussed in e.g Fettweis et al. (2020). The Greenland Ice Sheet SMB model intercomparison project (GrSMBMIP) suggested that discrepancies between RCMs are not systematic (Fettweis et al., 2020), thus there is a need for individual evaluation of each modeled melt volume product before we can compare the extent observed by ASCAT. We refer to Figure 5 for an overview of the evaluation taken prior to comparing RCMs with ASCAT. To establish a threshold (in mm w.e. day$^{-1}$) to infer the melt extent from the simulated RCM melt volume, we compare it to the PROMICE GC-net AWS. With this comparison we identify how much meltwater must be in the models before we can also observe it at the AWS. We compare each AWS to the RCM grid cell, which has the closest center point to the AWS location; see Figure 1 for AWS locations. It is important to note that the AWS measurements are significantly affected by local scale weather conditions, so this approach only ensures that the modeled melt by the RCMs aligns primarily at these specific locations. Therefore, we include the maximum number of AWS to align with, rather than being limited to those above the maximum snowline extent, which is where ASCAT and the inferred RCM melt extents are compared. Since melt is not directly measured at the AWS stations, we use 2m air temperature as a proxy for melt conditions, as near-surface air temperature is closely linked to melt processes. This approach allows us to identify and quantify temperature biases in each of the RCMs and assess how well the models simulate melt compared to in situ observations.

Air temperature is strongly correlated with melt since melt is a response to a positive surface energy balance, which occurs when the surface temperature reaches 0 °C (Cuffey and Paterson, 2010). However, it's important to note that air temperature and surface temperature are not the same; while air temperature influences surface conditions, surface temperature depends

on a combination of energy exchanges at the surface. Additionally, the local properties of the snowpack can also affect when melt occurs, and melt can occur in the snowpack when air temperatures are below 0 °C. Thus, it is crucial to consider that the heat content of the overlying atmosphere is not the sole driver of melt when we compare the modeled melt volume against the days with observed melt as indicated by AWS temperature measurements. With this approach, we apply a threshold to identify days with significant melt in the RCMs but also implement a threshold for the temperature observations to identify when melt occurs in the snowpack at the AWS. We explore various thresholds for temperature observations to account for other factors in the snowpack that influence when surface melt occurs.

A logical measure to identify how well the models and in situ observations align most closely would be to maximize the number of days where both the models and in situ observations agree on either melting or no melting. However, since the number of days with melt is fewer than those without melt, the dataset exhibits an imbalance. In the ablation zone, the imbalance is less pronounced with prolonged periods of melt in the summer, whereas in the accumulation zone, melt occurs for shorter periods. This data imbalance means that many days with agreement between RCMs and AWS can be attributed to seasonal patterns, concealing disagreement in the melt season. Instead, we utilize the receiver operating characteristic curve (ROC-curve, Peterson et al. (1954)) and the precision-recall curve (PR-curve Davis and Goadrich (2006)) to provide a more nuanced understanding of the alignment between RCMs and AWS. Given that $TP$ is the number of true positives, $TN$ is the number of true negatives, $FP$ is the number of false positives, and $FN$ is the number of false negatives, the ROC curve is a measure of the ability to distinguish between two classes across all thresholds and consists of a graph that shows the true positive rate $\left(TPR = \frac{TP}{TP+FN}\right)$ versus the false positive rate $\left(FPR = 1 - \frac{TN}{TN+FP}\right)$. Here, we define the AWS as true. Thus, we define the TP when the RCMs and AWS agree when melt is present, and TN is when no melt is observed both by the AWS and RCM. When melt is only observed at the AWS but not the RCM, it's defined as a FN and vice versa for FP. The ROC curve provides the total performance measure across all potential classification thresholds where a random model will produce a diagonal line, see Figure 5 as example. In contrast, a perfect model will have a ROC curve composed of the left and upper boundary lines. The goal is to choose a melting threshold that maximizes the TPR positives while minimizing the FPR. However, in an imbalanced data set, it is possible to produce a good ROC curve by making a large number of FP predictions, especially when the positive class is rare. Thus, for an imbalanced data set, it is important also to consider the PR curve, which is the fraction of TP among the positive predictions $\left(Precision = \frac{TP}{TP+FP}\right)$ versus the TP among the actual positives $\left(Recall = \frac{TP}{TP+FN}\right)$. When evaluating what melting threshold to apply to each RCM, we are looking for the RCM melting thresholds and AWS temperature threshold that maximizes the TPR positives while minimizing the FPR and maximizing the precision and recall.

Figure 6 shows the ROC-curves and PR-curves for possible RCM melting thresholds using different AWS temperature thresholds. The black dot indicates the optimal melting threshold for each RCM (Table 1). For HIRHAM5-ERA5 and HIRHAM5-ERAI, we see that both the best ROC-curve and best PR-curve match, although suggesting very different melting thresholds. For MARv3.12 and RACMO2.3p2 the ROC-curve suggests using temperature thresholds between -0.5 to 1 °C to find the best RCM melting threshold. However, the PR-curve suggests a lower temperature threshold between -1.0 to -2.0 °C, which yields better results. Table 1 showcases the applied thresholds informed by PROMICE temperature observations.

**Table 1.** Melting thresholds for the different RCMs based on in situ PROMICE AWS observations of air temperature. The table also gives examples of the mean July through August air temperatures at six selected AWS and the mean across all stations. The corresponding mean July through August air temperatures are showcased for each RCM. The locations of all stations included in the study are shown in Figure 1.

| | | Mean air temperature [°C] | | | | | | |
|---|---|---|---|---|---|---|---|---|
| | Thresholds [mm w.e. day$^{-1}$] | CP1 | DY2 | KAN_U | KPC_U | NUK_U | SDL | All AWS |
| **HIRHAM5-ERA5** | 4.1 | -6.36 | -6.78 | -5.08 | -1.13 | 0.74 | -9.62 | -4.02 |
| **HIRHAM5-ERAI** | 0.4 | -3.92 | -4.04 | -2.53 | 0.29 | 1.12 | -6.50 | -2.35 |
| **RACMO2.3p2** | 0.7 | -4.66 | -4.83 | -2.95 | -0.80 | 2.03 | -7.14 | -1.87 |
| **MARv3.12** | 1.0 | -5.16 | -5.14 | -3.31 | -0.93 | 1.24 | -7.71 | -2.64 |
| **PROMICE GC-net** | – | -4.96 | -4.70 | -3.31 | -0.85 | -0.84 | -6.80 | -3.39 |

To ensure consistency across datasets, all RCMs are regridded to a common grid, in this case, the ASCAT grid of 5.5 km resolution. For HIRHAM5 and RACMO2.3p2 we upscale data and apply the nearest neighbor interpolation method. For MARv3.12 downscale data and use a cubic interpolation method. It is important to note that regridding can potentially introduce a bias within the RCM output, meaning systematic errors not associated with internal parameterization choices within the RCMs. The potential implication of the regridding biases is considered when choosing a baseline threshold. Here, the aim is to apply a baseline threshold to all RCMs independent of warm/cold bias within the RCMs. Therefore, the baseline threshold was set to the smallest value possible (0.1 mm w.e. day$^{-1}$) without allowing regridding biases to impact the number of melt days. When comparing the RCM melt extent and ASCAT liquid water extent, we apply a similar snowline mask to the RCMs as ASCAT.

## 4 Results

Table 1 showcases the mean air temperature for July and August observed by AWS and modeled by the RCMs at the AWS stations, suggesting a cold bias in HIRHAM5-ERA5, whereas MARv3.12 and HIRHAM5-ERAI have a warm bias. However, the mean air temperature at selected stations highlights that the temperature bias is not systematic across the ice sheet. To evaluate the difference between the mean annual number of melt days in ASCAT and each RCM, we compute the RMSE for the whole ice sheet and each drainage basin (Rignot and Mouginot, 2012). Besides applying the in situ informed thresholds found in Tabel 1, we also apply the baseline threshold of 0.1 mm w.e. day$^{-1}$. Table 2 shows the mean annual number of melt days and the mean duration of the melt season for both the RCMs with the two thresholds applied and ASCAT. We define the start of the melt season when at least one grid point experiences melting. The melt extent using the in situ informed thresholds tends to align better with ASCAT observed mean number of melt days and the duration of the melt season. Furthermore, when the baseline threshold of 0.1 mm w.e. day$^{-1}$ is applied, melting occurs in parts of the SW basin all year. We apply a Mann-

**Table 2.** Mean annual melt days and mean duration of the melt season for each model using two different thresholds and for ASCAT. The melt season duration is defined as starting when at least one grid point experiences melting. A Mann-Whitney U test (Mann and Whitney, 1947) was applied to assess whether there is no effect of using an in situ-informed threshold. Additionally, the Rank-Biserial Correlation (r-value, Cureton (1956)) was computed to indicate the magnitude of the difference between thresholds.

| | Mean melt days | | Duration of melt season | | | |
|---|---|---|---|---|---|---|
| **Threshold method** | baseline | in situ | baseline | in situ | **p-value** | **r-value** |
| **HIRHAM5-ERA5** | 25 | 21 | 225 | 204 | > 0.001 | 0.48 |
| **HIRHAM5-ERAI** | 16 | 16 | 205 | 201 | > 0.001 | 0.49 |
| **RACMO2.3p2** | 24 | 18 | 365 | 344 | > 0.001 | 0.47 |
| **MARv3.12** | 25 | 18 | 274 | 166 | > 0.001 | 0.45 |
| **ASCAT** | 18 | | 154 | | - | - |

Whitney U test (Mann and Whitney, 1947) to test if there is no effect of using an in situ-informed threshold. The p-value in Table 2 suggests that it is very unlikely that there's no effect of using an in situ informed threshold.

The RMSE between ASCAT and RCMs are showcased in Table 3, where we see that applying the in situ informed threshold improves the discrepancies between modeled melt and ASCAT, except for HIRHAM5-ERAI. RACMO2.3p2 and MARv3.12
show the biggest improvement, although the improvement is not evenly distributed between drainage basins. Since the in situ informed thresholds generally reduce the differences between RCMs and ASCAT compared to the baseline threshold of 0.1 mm w.e. day$^{-1}$ (Table 3), we apply only the in situ informed thresholds in the following. A detailed comparison of the number of melt days using the baseline threshold is provided in Appendix A, showing larger discrepancies between ASCAT and RCMs, corresponding with findings in Table 3.

Figure 7 shows the spatial variability of the number of melt days both within each RCM (upper panel) and in comparison with ASCAT (lower panel). Looking at the spatial variability of the annual number of melt days for the RCMs and ASCAT (Fig. 4d), areas with >100 days of melt lie near the lower ablation zone, but the extent of areas with >100 days of melt varies from ASCAT and among models. Most noticeable is HIRHAM5-ERA5, which has substantially larger areas with >120 days of melt. Areas with <1 day of melt on average are shown in white in Fig. 7, illustrating areas where melt very rarely occurs.
Again, there is large variability among models in terms of modeling areas with almost no melt days.

While Table 3 illustrates the variability in error between ASCAT and RCMs across drainage basins, the lower panels in Fig. 7 reveal that the largest differences are concentrated near the ice margin across all basins. Generally, the SW and SE basins have the highest RMSE, explained by relatively large areas with 20 or more days of difference in these basins (Fig. 7). Similarly, HIRHAM5-ERA5 exhibits a high RMSE in the western basins, corresponding with large areas where HIRHAM-ERA5 models
20 or more melt days compared to what ASCAT observes. Although the in situ informed threshold reduces the differences, HIRHAM5-ERA5 continues to produce more melt days than ASCAT or any other model, even when the melting threshold is vastly greater than the other RCMs. On the other hand, HIRHAM5-ERAI consistently underestimates the number of melt days

**Table 3.** RMSE of the mean annual number of melt days modeled by RCMs using a baseline threshold and in situ informed thresholds and ASCAT across the whole ice sheet and for each Rignot and Mouginot (2012) drainage basins. RMSE is only computed inside the snowline data mask to mask out areas where ASCAT cannot detect liquid water; see Fig. 1.

| | HIRHAM5-ERA5 | | HIRHAM5-ERAI | | RACMO2.3p2 | | MARv3.12 | |
|---|---|---|---|---|---|---|---|---|
| **Threshold** | 0.1 | 4.1 | 0.1 | 0.4 | 0.1 | 0.7 | 0.1 | 1.0 |
| **Full ice sheet** | 10.70 | 7.67 | 7.39 | 7.76 | 28.77 | 6.27 | 17.78 | 4.93 |
| NW | 8.23 | 5.13 | 3.17 | 3.44 | 1.97 | 1.77 | 3.12 | 2.00 |
| CW | 9.01 | 5.80 | 3.30 | 3.74 | 2.10 | 2.10 | 3.11 | 2.65 |
| SW | 6.88 | 5.40 | 6.10 | 6.69 | 12.14 | 4.87 | 9.68 | 3.73 |
| SE | 5.65 | 4.89 | 5.91 | 6.07 | 31.56 | 5.78 | 18.75 | 4.08 |
| NE | 4.53 | 3.17 | 3.13 | 3.08 | 2.94 | 2.15 | 3.21 | 1.93 |
| NO | 5.11 | 3.35 | 2.49 | 2.68 | 1.56 | 1.38 | 1.66 | 1.76 |

across all drainage basins. MARv3.12 and RACMO2p2.3 show similar patterns of variability, where only large differences to ASCAT occur close to the maximum snowline elevation.

Results from all RCMs and ASCAT (Fig. 8a) indicate that, on average, the melting season begins at the beginning of May and culminates around July when the greatest melt extent occurs. While RCMs suggest that the melt season, on average, ends around mid-September, small melt areas are still observed in ASCAT well into October. The maximum melt extent is, on average, approximately 30 % of the ice sheet, except for HIRHAM5-ERA5 with >35 %. At the beginning of the melt season, ASCAT detects the increase in the extent of liquid water 10-15 days earlier compared to when the RCMs simulate an increase in the melt extent. However, the decrease in the extent of liquid water at the end of the melt season corresponds well with the modeled melt extent. In August and at the beginning of September, the melt extent decreases but with small periodical increases, which ASCAT also detects prematurely, although not as pronounced as at the beginning of the melt season.

On 12. July 2012, an extreme melting event was observed across almost the entire ice sheet (Nghiem et al., 2012). We can use this year to showcase how well an extreme but relatively short melt event is captured in the RCMs and by ASCAT. Results show that using the in situ informed thresholds, only RACMO2p2.3 simulates melting of >90% of the ice sheet, but the rest only predict approximately 80% (Fig. 8b). Figure 8b also shows that liquid water is detected earlier by ASCAT at the start of the season. During the increase in melt extent from July to mid-August, ASCAT was able to capture an increase in melt extent, but the magnitude of the increase was smaller compared to the models.

## 5 Discussion

The goal of this study is to use ASCAT to understand how RCMs model ice-sheet-wide melt extent compared to observations and further pinpoint biases within models leading to the observed discrepancies. By comparing RCM with PROMICE GC-net

temperature observations, we are able to determine how well the RCMs align with in situ observations. While the uneven distribution of AWS stations, with a concentration in the ablation zone, may result in certain areas being better represented in the assessment than regions with fewer AWS stations, Tab. 3 shows that by ensuring that the RCMs align with in situ

measurements at specific locations, the modeled melt extent and satellite observed melt extent show better agreement. Further, with this approach, we identify that HIRHAM5-ERA5 shows large discrepancies from PROMICE GC-net air temperature observations. The melt threshold in HIRHAM5-ERA5 is considerably higher than the remaining melt estimates, indicating that HIRHAM-ERA5 overestimates melt. The identified melting threshold of 4.1 mm w.e. day$^{-1}$ suggests potential issues with the representation of melt in HIRHAM5 when forced with ERA5. Consequently, this highlights the importance of carefully

considering model performance. When we compare the number of melt days in the RCMs versus those observed by ASCAT, MARv3.12 has the lowest RMSE for the whole ice sheet. However, MARv3.12 performs predominantly better on the eastern side of the ice sheet, whereas RACMO2p2.3 performs better on the west coast and in the northernmost areas. While none of the RCMs exhibits perfect agreement with either PROMICE GC-net or ASCAT observations when looking at the RMSE and the spatial differences between the annual number of melt days, neither MARv3.12 nor RACMO2p2.3 has an advantage

over the other. On the other hand, HIRHAM5-ERA5 and HIRHAM5-ERAI display considerable discrepancies in terms of number of melt days when compared to ASCAT, each displaying unique patterns of deviation. Our analysis shows the value in using common independent spatially and temporally varying datasets to evaluate and improve models and we suggest that this ASCAT dataset is a useful addition to better understand biases in models, especially as it is independent of other datasets such as MODIS that have been used to develop models.

## 5.1 Biases in RCMs

To get the most valid comparison between each RCM and ASCAT, we utilize in situ observations to assess biases and to determine an appropriate threshold for the melt extent in RCMs. By fitting each RCM to in situ observations we minimize the differences that are introduced due to model set-ups like resolution, parameterization etc. Thus, we reduce overall inter-model discrepancies as well as differences in melt extent compared to that observed by ASCAT. Despite applying the in situ informed

thresholds, persistent patterns between RCMs and ASCAT remain. HIRHAM5-ERA5 is the only model that, on average, predicts more melt days than ASCAT while also having a substantially higher melting threshold compared to the other RCMs melt outputs, see Tabel 1 and 2. Using the in situ informed thresholds RACMO2.3p2, and MARv3.12 model the same mean number of melt days as ASCAT, while HIRHAM5-ERAI models slightly fewer melt days. Looking at Figure 7 HIRHAM5-ERAI, RACMO2.3p2, and MARv3.12 all model fewer melt days than ASCAT on average in the lower accumulation zone,

indicating either a limitation in the observation of melt by ASCAT or that melt is not well represented in RCMs in this area. However, the magnitude of disagreement between RCM and ASCAT varies from RCM to RCM and region to region (Table 3), again suggesting that the melt and melt extent in these areas is not well represented in the RCMs, or that the melt threshold should be considered non-stationary according to local conditions. While HIRHAM-ERA5 exhibits the largest meltwater production (Fig. 2), HIRHAM-ERA5 is also characterized by the lowest mean JJA air temperatures (Fig. 3a-d).

In general, HIRHAM-ERA5, MARv3.12, and RACMO2.3p2, all forced with the ERA5 reanalysis, tend to report lower air

temperatures compared to HIRHAM5-ERAI. This trend is likely originating from ERA5 exhibiting lower air temperatures than its predecessor, ERAI, as highlighted by (Krebs-Kanzow et al., 2023). While some variability among the models forced with ERA5 exists, the models commonly exhibit a higher mean JJA downward shortwave radiation at the surface (SWD) compared to HIRHAM-ERAI. Krebs-Kanzow et al. (2023) similarly reported an overestimation of SWD in ERA5 compared to

345 ERAI. Since the differences in air temperature and SWD between ERA5 and ERAI have implications for meltwater production and ultimately runoff, Krebs-Kanzow et al. (2023) concluded that replacing ERAI with ERA5 forcing in an energy balance model of the Greenland Ice Sheet requires some recalibration to reproduce existing observations. In this study, we show the implications on meltwater production and extent when forcing the HIRHAM5 SMB model with ERA5 instead of ERAI without recalibration, though the inclusion of a different albedo scheme is likely to be more important.

In addition to the distinct forcing fields in the two HIRHAM5 outputs, the approach for determining surface albedo differs. The simple surface albedo scheme in HIRHAM-ERA5 results in a lower mean JJA surface albedo in the ablation zone and lower accumulation zone compared to HIRHAM-ERAI, which included MODIS observations in the surface albedo scheme (Fig. 3e and 3f). RACMO2.3p2 is characterized by the lowest surface albedo across the entire ice sheet, while MARv3.12 and HIRHAM5-ERAI are dominated by lower surface albedo, especially in the accumulation zone. Surprisingly,

while RACMO2.3p2 and HIRHAM5-ERAI both report to incorporate MODIS bare ice albedo observations in the surface albedo computation in the ablation zone, the resulting surface albedo differs greatly between models. In the RCMs, the surface albedo is a crucial parameter for simulating the surface energy balance, contributing to a higher surface energy balance when the surface albedo is low. A low albedo in the ablation zone and lower accumulation will increase the meltwater production in these areas. Due to the high variability of air temperature, SWD, and surface albedo among models, the albedo parametrization

and radiation and temperature schemes within models should be critically assessed using high quality measurements to reduce the observed discrepancies in the models' estimates of meltwater production (Fig. 2). By aligning simulated melt rates more closely with observational data, we can improve the model estimates of meltwater production and ultimately runoff.

## 5.2 Limitations of ASCAT melt observations

We use the maximum extent of the snow line to mask out the ablation zone when comparing the RCMs to ASCAT due to

365 nonphysical observations of refreezing by ASCAT in these areas (Fig. 7). This is not only due to biases in the melt classification algorithm but also comes from the microwave data itself. Although melt causes a drop in the backscatter signal, processes other than the refreezing of liquid water can counteract the decrease in the backscatter signal. Once the surface starts to melt, the surface roughness changes, which can cause an increase in the backscatter signal, potentially concealing the liquid water and melting in the backscatter signal. Since the effect of changing surface roughness on the backscatter signal is not fully

understood, it is difficult to remove this bias in liquid water detection other than removing areas where changes in surface roughness are very pronounced. We further see this systematic misclassification of refreeze or no melting in ASCAT when investigating the 2012 melt season. Here, ASCAT detects a lower liquid water extent in the late melt season compared to the melt extent simulated by the RCMs. Since the refreezing periods identified from ASCAT data are not included in the

melt season analysis, misclassifications that stem from melting being misclassified as refreezing due to change in the surface
roughness are not included, potentially contributing to a smaller melt extent at the end of the melt season.

Detectable in both the 2012 melt season (Figure 8b) and the annual mean cycle of melt extent (Figure 8a), ASCAT detects
the increase of liquid water extent before the RCMs simulates the increase in melt extent, partly explained by the preprocessing
averaging done to enhance the spatial resolution. However, based on the Figure 7 we asses that on average ASCAT detects the
liquid water more than 10-15 days before the the RCMs simulates melt, meaning that the precessing averaging cannot fully
explain the differences between ASCAT and RCMs. Although most pronounced at the beginning of the season, we see that
the earlier liquid water detection also occurs when there's an increase in the melt extent in the late season. On average, the
magnitude of the melt seasonal cycle of melt extent agrees well with RCMs, suggesting that liquid water is observed earlier
but at similar extent.

Finally, the ASCAT backscatter varies due to additional factors such as specific properties of the snow, e.g. grain size and
the presence of ice, due to its influence on the dielectric properties and roughness geometries. Here, two possibilities consist in
progressing the melt retrievals of ASCAT; we could use the surface properties from the individual RCMs or in situ observations.
For the latter, there is a seasonal bias in the in situ observations and a lack of spatial coverage, making this difficult to use for
ice-sheet-wide earth observation data production. As for the RCMs implementation, this would hamper the ASCAT data as an
independent data record.

## 6  Conclusions

ASCAT provides a valuable tool for evaluating the performance of RCMs, which is currently the only source for assessing
the melt volume on a global scale of the ice sheet. By utilizing observations of melt extent by ASCAT, we can evaluate
how well four different RCMs' melt outputs represent melt spatially and temporally across the Greenland ice sheet at the
present day. To ensure that the RCM-modeled melt aligns well with in situ observations, we compare the RCM melt output at
PROMICE GC-net AWS. Assuming a strong correlation between 2m air temperature and melting, we can use PROMICE air
temperature measurements to assess temperature biases and determine a melting threshold in the RCMs to identify days with
significant melt. Here, we find that HIRHAM5 forced with ERA5 shows potential issues with the representation of melt due to
its relatively poor alignment with PROMICE air temperature measurements. HIRHAM5 forced with ERA-Interim, MARv3.12,
and RACMO2p2.3 all show similar alignment with PROMICE, but when comparing with ASCAT melt extent, MARv3.12 and
RACMO2p2.3 show better agreement with ASCAT, but each RCM shows distinct patterns of variations. This suggests that
in some cases one RCM can represent present-day melt extent across the entire ice sheet more effectively than the ensemble
mean, as variability among models does not appear to be random variations. For all RCMs included in this study, the onset of
melt occurs more than a week later than what is observed from ASCAT, even when accounting for the averaging of satellite
data. Further, we see that the regions with the largest differences in total number of melt days across all RCMs are the SW and
SE basins of the ice sheet, indication that melt is likely not well represented in the RCMs in these areas.

The variability among modeled JJA air temperature, SWD, and surface albedo can explain the large discrepancies among modeled meltwater production (Fig. 2). Notably, we show the implications for meltwater production when running the HIRHAM5 SMB model using ERA5 instead of ERAI without any recalibration. Further, by ensuring that the models accurately simulate the variability in albedo, such as through incorporating MODIS bare ice data, it can lead to a more accurate representation of the surface energy balance, and consequently, meltwater production. Despite both RACMO2.3p3 and HIRHAM5-ERAI using similar MODIS bare ice observations and having similar dynamical schemes, the surface albedo and its effects on meltwater production varied substantially between models, highlighting the need for a critical evaluation of model outputs against high-quality measurements to reduce inter-model discrepancies. In general, our analysis demonstrates the value of using independent datasets like ASCAT to identify the spatiotemporal variability of RCM simulated melt. This approach complements traditional model validation methods and intercomparison exercises, which can inform future model development to better simulate ice sheet surface melt and possibly be incorporated in future MIP efforts.

*Code availability.* Code is publicly available at https://github.com/anpug/4DG_meltmaps.git

*Data availability.* ASCAT surface melt extent maps are available at https://cryoportal.enveo.at/data/ under meltmap. PROMICE GC-Net AWS 2m air temperature are available on GEUS Dataverse (https://doi.org/10.22008/FK2/IW73UU, How et al. (2022)). Further, historical GC-net 2m air temperature data is available on GEUS Dataverse (https://doi.org/10.22008/FK2/VVXGUT, Steffen et al. (2022)). HIRHAM5 forced with ERA-Interrim is available at https://ensemblesrt3.dmi.dk/data/prudence/temp/nichan/Daily2D_GrIS/ (Langen et al., 2017) and HIRHAM5 forced with ERA5 will be made availble once the paper is published https://doi.org/10.11583/DTU.25568040 . RACMO3.2p2 is freely available upon request (Noël et al., 2019). MARv3.12 is available at ftp.climato.be/fettweis/MARv3.12/Greenland (accessed May 2023, (Tedesco and Fettweis, 2020))

**Appendix A: Comparison between RCMs and ASCAT using the baseline threshold of 0.1 mm w.e. day$^{-1}$**

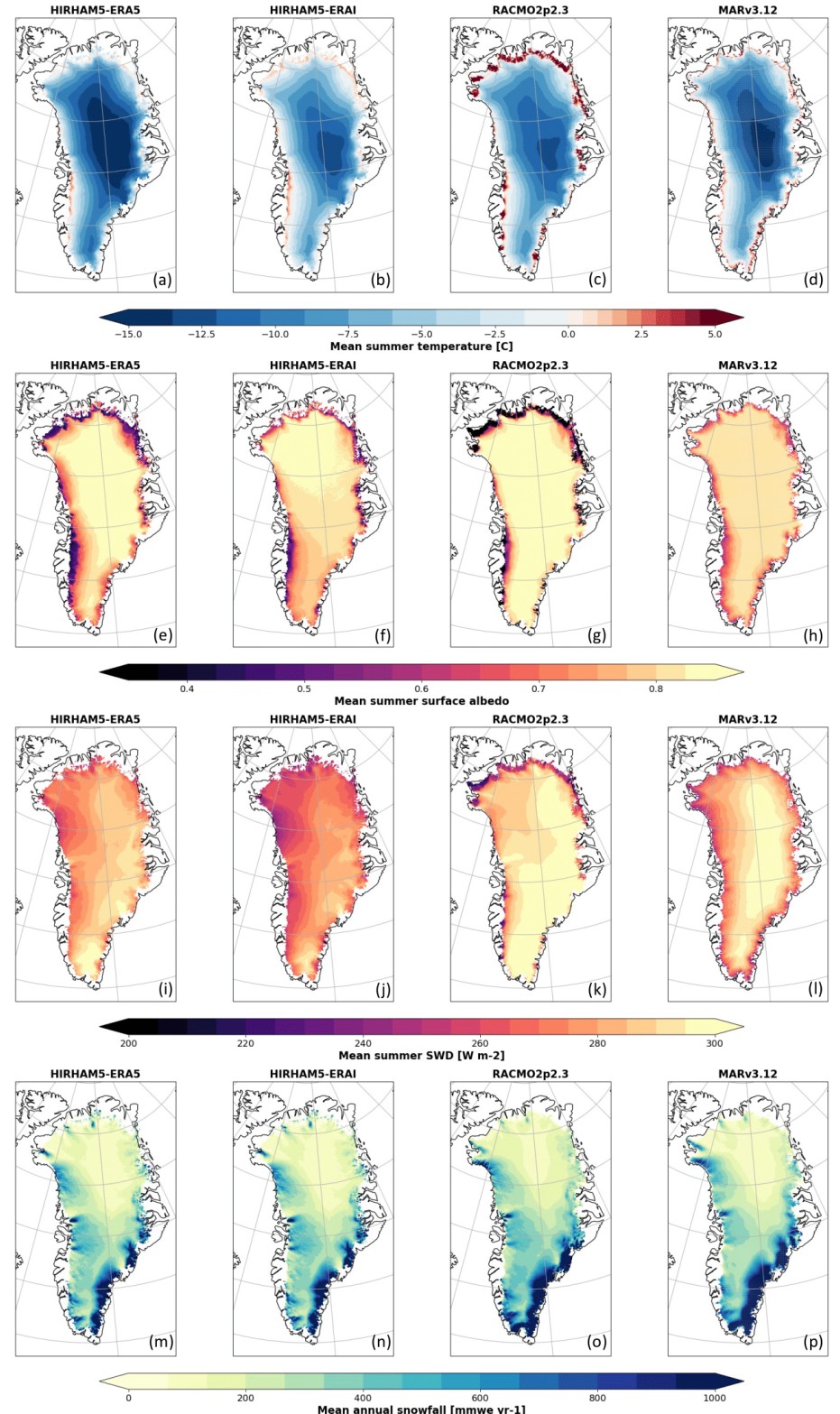

**Figure 3.** RCM simulated varibles from 2007-2020 for HIRHAM-ERA5, HIRHAM-ERAI, MARv3.12 and RACMO2.3p2. (a-d): mean JJA temperatures in °C, (e-f): the JJA surface albedo, (i-l): the JJA SWD in W m-2, and (m-p): the mean annual snowfall in mm w.e. yr$^{-1}$.

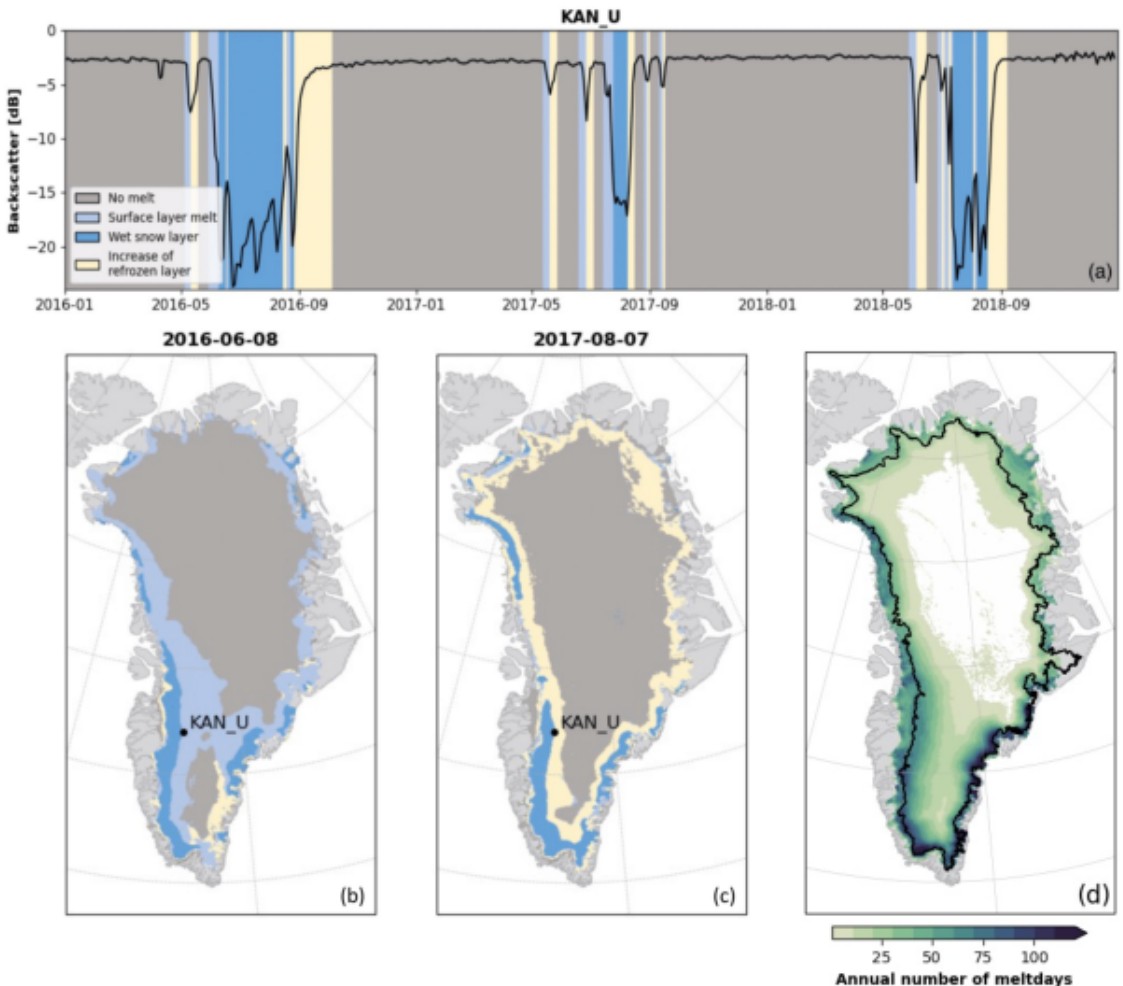

**Figure 4.** (a) 2016-2018 melt detection and the backscatter signal at KAN_U. (b-c) Two examples of the spatial distribution of liquid water detection on the 8[th] of June 2016 and 7[th] of August 2017. (d) Mean annual number of days with detected liquid water, but without refreezing. In (d) we apply the maximum elevation of the snowline between 2007-2020 (black line) to mask out areas where ASCAT persistently cannot correctly detect the presence of liquid water.

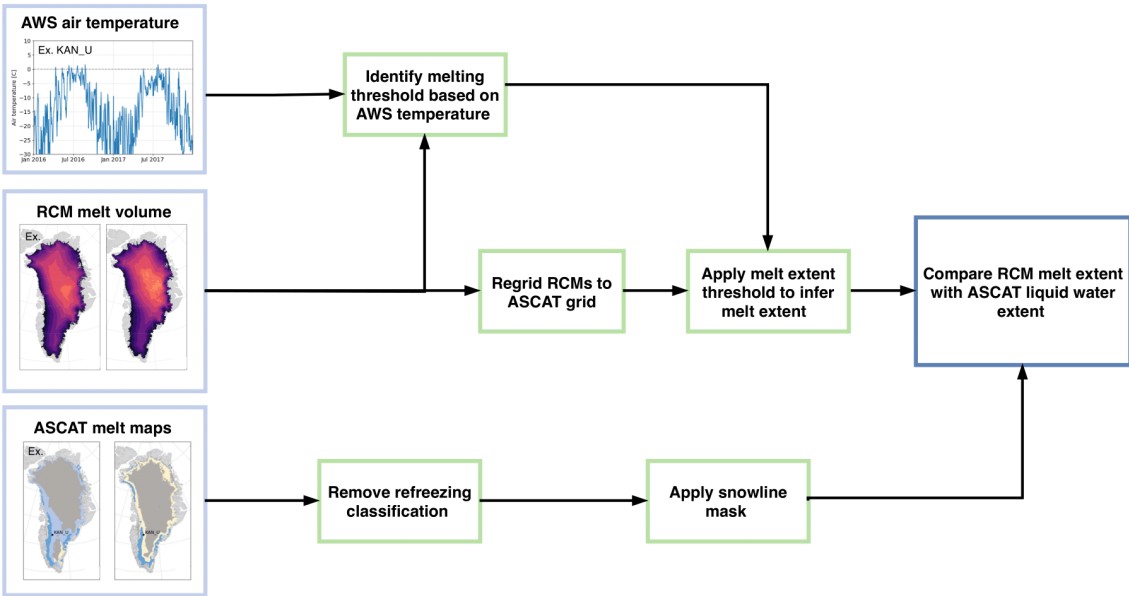

**Figure 5.** Diagram describing the evaluation process of RCMs melt volume and processing of ASCAT surface melt extent maps prior to the comparison between ASCAT liquid water extent and RCM inferred melt extent.

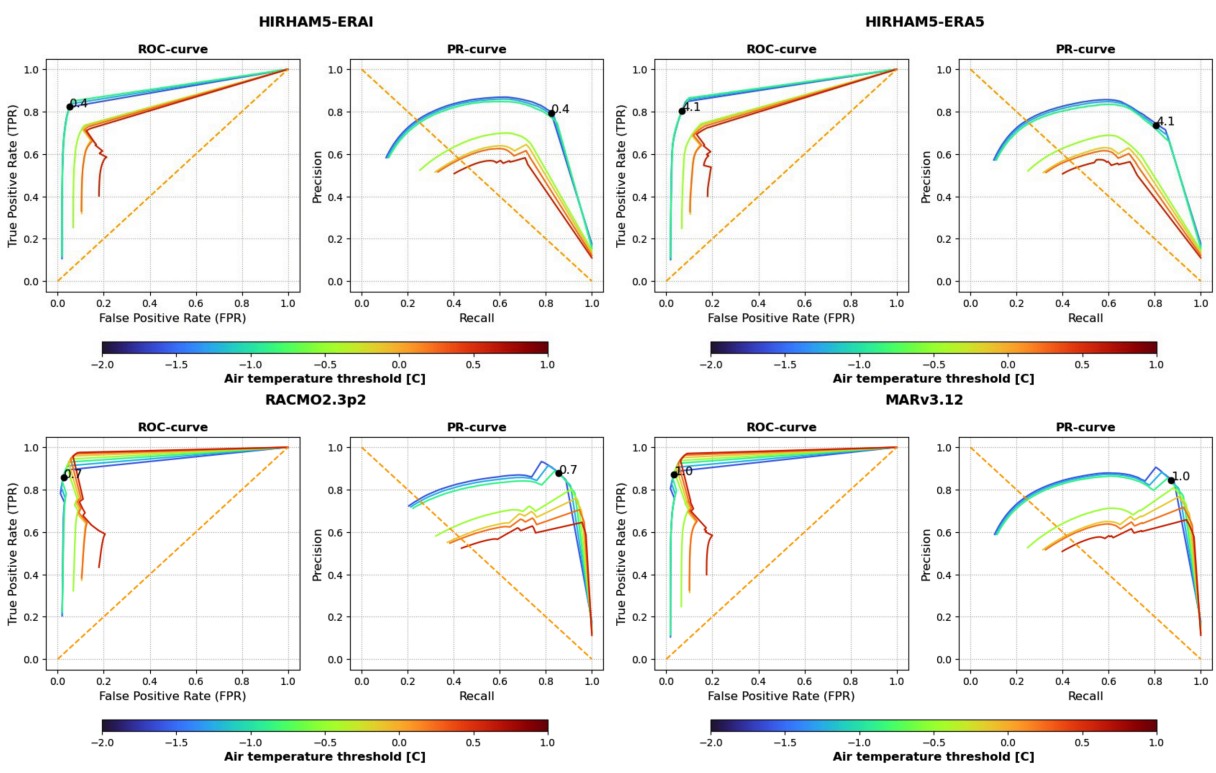

**Figure 6.** ROC- and PR-curves for possible RCM melting thresholds using varing AWS temperature thresholds. The black dot indicates the optimal melting threshold for each RCM given in Table 1.

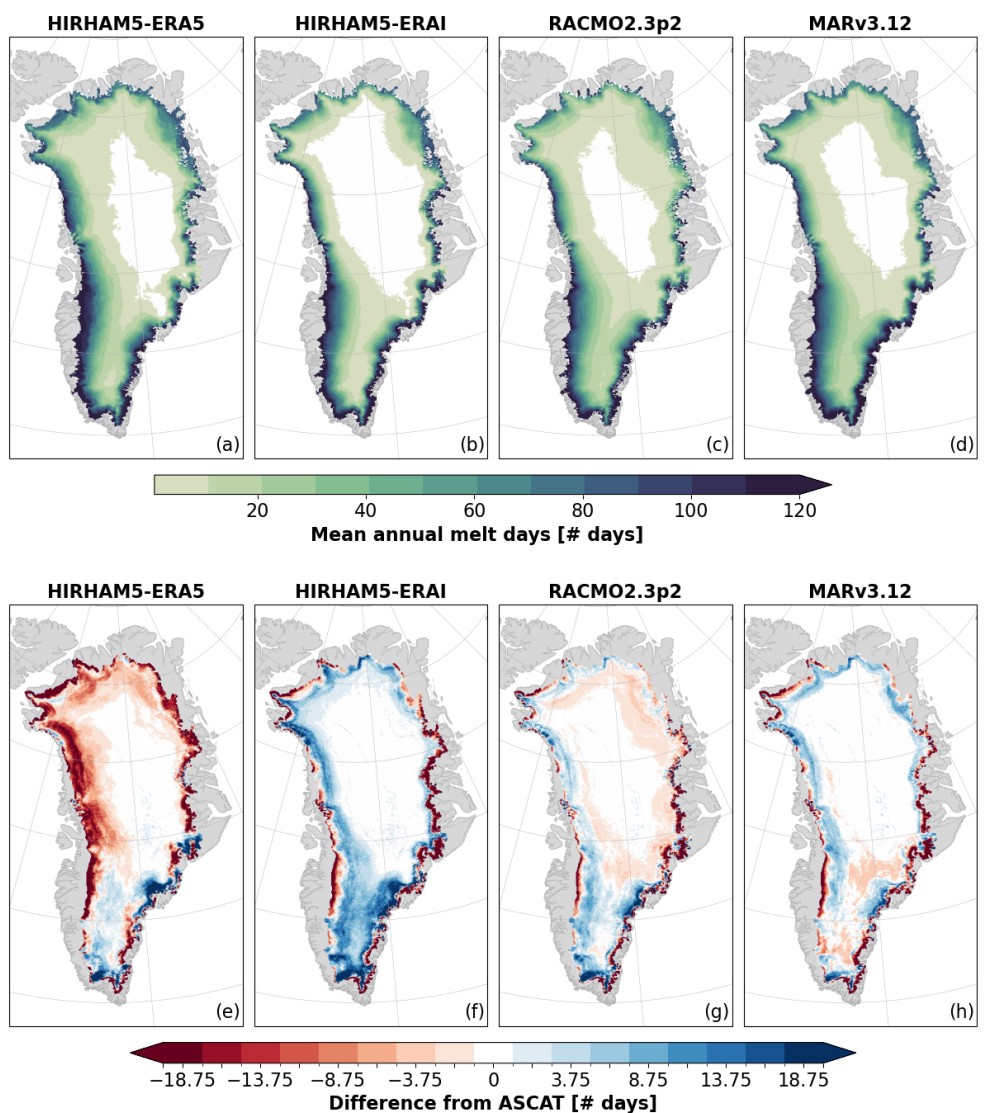

**Figure 7.** (a-d): The mean annual number of melt days modeled by the RCMs using an in situ informed melt threshold to defined days with significant melt. Pixels with <1 day of melt on average are marked as white, showcasing areas where melt rarely occurs. (e-h) The mean annual difference between the number of melt days in ASCAT and RCMs areas above the 2007-2020 maximum snowline elevation (Fig. 4d). Red areas correspond to more melt days in ASCAT on average and blue areas correspond to more melt days in the RCM on average. Melt in ASCAT is defined as Label ST-2A and ST-2B.

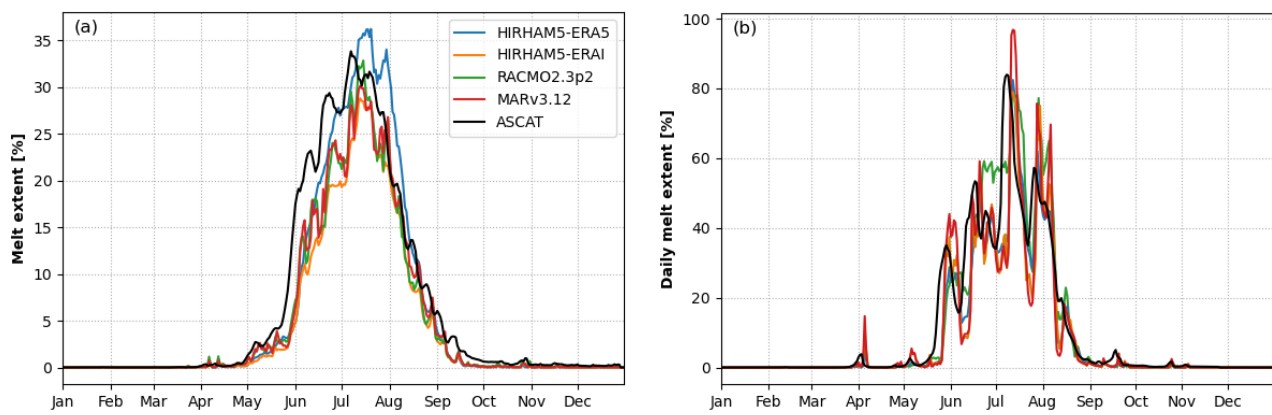

**Figure 8.** (a) Mean seasonal melt extent [%] modeled by RCMs using in situ informed thresholds and retrieved from ASCAT (2007-2020). In situ informed thresholds are given in Table 3. (b) Daily melt extent [%] modeled by RCMs using in situ informed thresholds and retrieved from ASCAT in 2012. In situ informed thresholds are given in Table 3. The melt extent in both (a) and (b) is constrained to areas above the maximum elevation of the snowline.

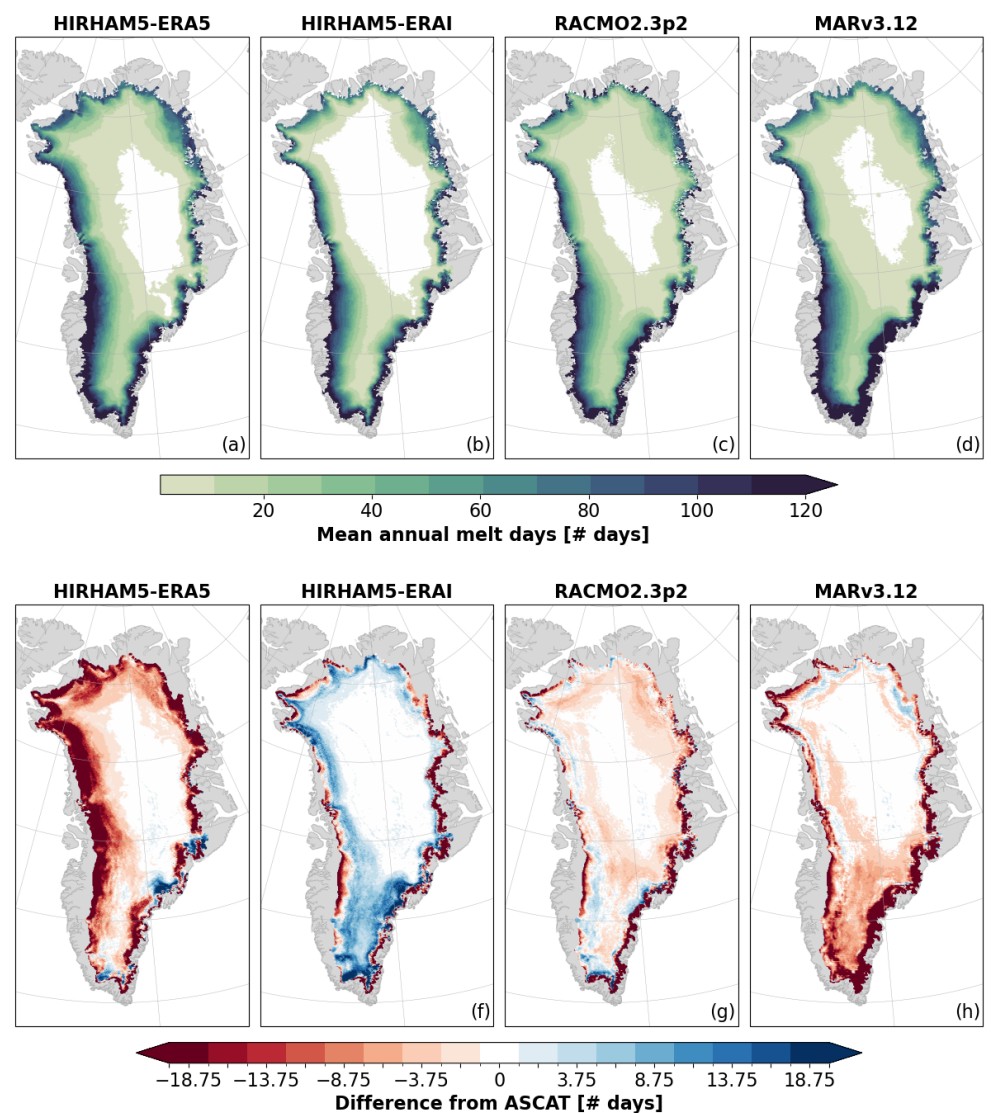

**Figure A1.** (a-d): The mean annual number of melt days modeled by the RCMs using a baseline threshold (0.1 mm w.e. day$^{-1}$) to define days with significant melt. Pixels with <1 day of melt on average are marked as white, showcasing areas where melt rarely occurs. (e-h) The mean annual difference between the number of melt days in ASCAT and RCMs areas above the 2007-2020 maximum snowline elevation (Fig. 4d). Red areas correspond to more melt days in ASCAT on average and blue areas correspond to more melt days in the RCM on average. Melt in ASCAT is defined as Label ST-2A and ST-2B.

*Author contributions.* AS and LSS framed the original study, AP further developed this and wrote the first draft of the paper with the help of AS, NH, LSS, RM, and SBS. TN, SS, and JW developed the presented ASCAT classification algorithm and provided support for the data analysis and discussion. NH processed and provided the HIRHAM5 data. All authors have revised the manuscript.

*Competing interests.* At least one of the (co-)authors is a member of the editorial board of The Cryosphere.

*Acknowledgements.* The authors thank the editor and three anonymous reviewers for constructive comments and feedback. AP, NH and RM are supported by the Danish State through the National Centre for Climate Research (NCKF). Furthermore, NH is supported by the Novo Nordisk Foundation project PRECISE (NNF23OC0081251). Model simulations and analysis in this publication were supported by the project PROTECT funded by the European Union's Horizon 2020 research and innovation programme under grant agreement No. 869304. TN, JW, AS and SS acknowledge support from the European Space Agency through the POLAR+ 4DGreenland (ESA Contract No. 4000132139/20/I-435 EF) and 4DAntarctica project (ESA Contract No.801ESA/AO/1-9570/18/I-DT). Further, AS is supported by The Programme for Monitoring of the Greenland Ice Sheet (PROMICE).

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
