# Peer review of "Bias in modeled Greenland ice sheet melt revealed by ASCAT"

_EGUsphere, 2024_

## Referee Comment (RC1)

A Review of: "*Bias in modeled Greenland ice sheet melt revealed by ASCAT*"
Puggaard *et al.* 2024, The Cryosphere Discussion.

Purggaard and co-authors compare here extent of meltwater production over Greenland as represented by 3 different RCMs to a remote sensed product from ASCAT instrument. Their aim was to identify differences between these 3 RCMs. To realize this comparison, they needed to determine a threshold of melt in the model which append before melt events are actually detected at the surface which infer the melt extent in the RCMs. To do so, they compare melt rate from the different models to in-situ observation of the near-surface temperature from AWS (PROMICE and GC-Net networks). As these thresholds are adapted to each model, discrepancies between ASCAT product and RCMs, as well as between models, are reduced (compared to a fixed threshold common to all models).

Main conclusions are

- biases between the RCMs are coming from: different albedo schemes, representation of the snowfall, and temperature as well as differences in radiation schemes;
- The ASCAT product presents some limitations (mismatch in the ablation area of the ice sheet, bias in the melt season temporality);
- RCMs present later melt season, and earlier end of this season than ASCAT;
- Slightly smaller melt area in RCMs than in ASCAT.

**General comments**

The manuscript is generally well written and easy to read. The originality and interest of this study come from the comparison to several modeled melt products. I will have several major comments to improve the quality and the consistency of the manuscript.

In the method, different parts are poorly described which not deserve the study. More the way you build your datasets, and your tools are described, better is the clarity of the method, specifically if external people to remote sensing community is reading your work. These parts are highlighted with some specific comments in the next section and should not be difficult to address.

It seems to me to be rather poorly considered that the data derived from ASCAT don't give an indication of meltwater in particular, but of all the water present on the surface and in the first layers of the snowpack. This is well presented in the introduction, but in some other parts of your text, this is a bit more confusing. Does it make sense, then, to compare this extent with melt extent in the models? And not the liquid water content in the first snow layers (see for instance Dethinne *et al.* 2023 and Picard *et al.*, 2022)? Are we sure that the variables selected in the models are indeed representative of what the satellite observes? From another point of view, if we look at the problem the other way round, we

could ask ourselves whether an additional ASCAT data processing step could be applied to translate them more specifically into meltwater.

The comparison between RCMs is original for this type of comparison (not included in a "MIP" exercise) with remote sensed data but is also audacious and complicated. This kind of work require deep knowledge of how processes inside the model are modelized/parametrized to be able to compare it with other products. I recommend to deeply double check the way that components of meltwater production or water in the snowpack are determine in the 3 different models and be sure that everything is comparable. In that aspect, not including MAR and RACMO people is a risky move, or at least limit the study to HIRHAM alone. The idea is not to transform this study into an intercomparison of these 3 models, but to better understand it to highlight why the biases you mention are present.

**Specific comments**

**Data**

L119: For MAR add the same detail-level than the 2 other models (at least how albedo is retrieved).

L136-137: Could you a bit more detail the SIR algorithm? As you need more measurements, is the reconstruction constant in time and in space? Are there any supplementary uncertainties bring with this method?

**Methods**

- L170-171 : "We compare the RCM output of surface melt with observed 2 m temperature data", even if it could be obvious, if think this kind of sentence could be not so easy to understand at first read. I suggest switching some parts of the second paragraph of your method with the first one to be more readable.

- Could you more detail how you construct the ROC- and PR-curves. Specifically, how do you determine if it is a true or false melt-day compared to AWS if you already calculate it for different temperature rate. If I understand well, you don't have directly melt rate observed at the AWS, but a guessing relates to the temperature measured, transform to melt through a lapse rate correction (which should be explained here). I think something is unclear for me here because of a lack of details.

- Please, explain here your method to choose the grid cell(s) corresponding to your AWSs. Is it the nearest neighbor, or the 4-nearest, ...?

- L195: You should reefer earlier to Fig. 5 to illustrate how understand the ROC- and PR-curves.

- L207: You're talking about a threshold of -2°C (also in Table 1), but in your Figure 5, your curves are only from -1 to 1 °C. How did you determine this threshold?

- Table 1: Why only one melt threshold and one per AWS for temperature? Is it an average from all melt rate retrieved at each AWS? Your method needs to be better described for that too. Moreover, are you only using these few AWSs as presented in the Table 1? Why only these ones? In the AWS description section (2.1), you mentioned that you will use all the AWS, at least much more than presented in Table 1.

- Still concerning observation from AWS, how do you manage the fact that most of the AWSs are in ablation area, and probably in the area where ASCAT cannot correctly detect the presence of liquid water, that you even mask outed? Or do you only consider AWSs inside the ASCAT mask? Otherwise, you decide to correct RCMs' melt with a threshold determine with comparison outside your area of melt comparison, which is not 100% valuable. Also, could please consider adding this mask in your first figure to well situated it compared to AWSs' localization as well as a more detailed comment on how you retrieved it.

**Results**

- L242-243: "ASCAT detects the increase in melt extent earlier compared to RCMs" Isn't it due to the detection of water by satellite and not directly melt (cf. 2nd general comment)?

- L249: can we talk about 'prediction' here as RACMO is prescribed by reanalyses at its lateral boundaries? Please rephrase with another verb.

- Table 2: Could you add the mean number of melt day for both observation and RCMs to compare your RMSE. You should also compare the difference between your two methods (uniform or in situ informed threshold) to determine if the gain with one or the other is significant (or real statistical test, it's even better).

**Discussion**

- L258-259: "Tab. 2 shows that by ensuring that the RCMs align with in situ measurements at specific locations." You cannot claim that RCMs align with measurement only by considering the RMSE. You need deeper statistical analyze to claim this.

- L274-276: Apply different melt threshold (based on the same in situ observation) for the different RCMs is also revealing a certain kind of bias in the model. Could you discuss that too in your Discussion?

- L297-298: First you say that RACMO present the lowest albedo, then you also explain that MAR and HIRAM have a lower albedo. Could you rephrase to better emphasize what are the differences and key features for each model/group of models?

- L300: Could you investigate a bit more why you have such differences in RACMO and HIRHAM MODIS-based albedo in the ablation area? If it's possible, it could be nice too also compare the different albedo of the models to the MODIS albedo.

- You do not talk about the differences on how the models represent the firn layer, whereas in you introduction you mention that "*The magnitude of the decrease in backscatter varies due to factors such as the snow water content and the specific properties of the snow, such as grain size and the presence of ice layers and lenses, which influence the dielectric properties and roughness geometries (Wismann, 2000; Long, 2017).* " I heard there that the signal to detect (melt)water at the surface is dependent of the snowpack conditions which are not represented/modelled in the same way in the 3 models. It should be a supplementary discussion point as melt event, and presence of water, could be delay, or more or less important, due these different way to model/parametrize the firn layers, then lead to difference when compared to ASCAT.

- Figure 3 a-d: center your color bar to 0, it's misleading as it is. And please use only 2 varying colors, one for positive and the other for negative values. Also, please avoid yellow at pivotal value.

- Figure 3, RCMs' Albedo: concerning the MAR model, you plot albedo for entire land areas and not only what looks like an ice mask in the 2 other models. Are you sure you plot the albedo used in the melt calculation, meaning the one for the ice grid points? Concerning the albedo from RACMO, considering the intercomparison and preliminary feedbacks from the PROTECT project, the albedo from RACMO presented here suggests high values, hinting a potential error when choosing which albedo plot.

**Conclusions**

- L340: "[...] *can lead to more accurate simulations of surface energy balance.*" Could you rephrase, as you don't actually look at the entire surface energy balance, but only some components.

**Appendix**

Figure A1 is exactly the same than Figure 6. Is it necessary as the appendix are in the continuity of the text and not in another document as Supplements?

*Technical corrections*

- L109 ACMO2.3p → RACMO2.3p2;
- L111 2x in a row "On the lateral boundary,";
- L115 2x "." in a row;
- L144-145 : 2 times in a row : "the first and second";
- - Caption of Table 1: There is something wrong in this sentence: "Melting thresholds for the different RCMs based on in situ PROMICE AWS observations of 2m temperature and mean air temperature for August and July simulated by the RCMs at AWS stations and observed by the AWS stations using a lapse rate correction." I think you need to remove 'and observed by the AWS stations'.
- L232: close the bracket here: "(Fig. 6.";

- L357: HIMHAM5 data → HIRHAM5 data.

**References**

Dethinne, T., Glaude, Q., Picard, G., Kittel, C., Alexander, P., Orban, A., & Fettweis, X. (2023). Sensitivity of the MAR regional climate model snowpack to the parameterization of the assimilation of satellite-derived wet-snow masks on the Antarctic Peninsula. The Cryosphere, 17(10), 4267-4288.

Picard, G., Leduc-Leballeur, M., Banwell, A. F., Brucker, L., & Macelloni, G. (2022). The sensitivity of satellite microwave observations to liquid water in the Antarctic snowpack. The Cryosphere, 16(12), 5061-5083.

---

## Referee Comment (RC2)

Review of "**Bias in modeled Greenland ice sheet melt revealed by ASCAT**" by Anna Puggaard and co-authors

This manuscript combines a novel processing of scatterometer data (ASCAT) with regional climate model (RCM) data to investigate the melt extent above the snowline of the Greenland Ice Sheet (GrIS). ASCAT data are here used as a reference to evaluate differences between four different RCM derived melt products. Additionally datasets are "aligned" to observations using near-surface air temperature observations from automatic weather stations (AWS) on the GrIS. The authors find distinct differences between the different RCMs with respect to the melt extent and the number of melt days. This study is of great interest to the community as it demonstrates that satellite data might provide a valuable reference to evaluate melt products for regions with little to moderate melt. With rising temperatures these regions might become increasingly important in the near future. The manuscript is generally well written but sometimes it lacks attention to detail. The objectives of the study could be accentuated more. Also the chosen strategy how to "align" observations and model data is poorly motivated.
In summary I recommend major revisions before publication.

**General remarks:**

Essential pieces of information are missing in the method part (in some cases these are implicitly given in the text): which period is taken into account, what frequency is analysed (hourly, daily, other?), how is different spatial and temporal resolution in the different data sets treated? How are datasets regridded? There are regridding biases mentioned- these could be illustrated or estimated.

In my view the introduction could point out more clearly the potential benefit of the ASCAT data set: Surface mass balance estimates have been mostly evaluated and also tuned with respect to the (basin wide) mass balance of the GrIS. Potential biases in melt rates above the snow line might be overlooked like that, as these are not necessarily resulting in mass changes. However these regions, where melt occurs only sporadically today, might turn into regions which contribute to sea-level rise in the near future.

It seems overambitious to try to investigate why different RCMs represent melt rates differently without a much deeper dive into characteristics of snow properties and climate forcing. On the other hand, to show that the different models simulate different melt extents does not require a satellite data set.
Instead I would propose to emphasize and focus on questions like:
Is the onset of melt detected too late systematically? How many days?
Are there differences between regions which experience surface melt every year and regions where melt occurs only in extreme melt years like 2012.
Is the length of melt periods overestimated by ASCAT due to the residual meltwater in the snow pack? If so, is this bias particularly pronounced for long, intense or short periods? Where melt periods occur intermittently within one year, are later melt periods represented differently than early melt periods (idea: the albedo might not recover fully after a melt event and the snow surface might be more vulnerable- which is potentially not represented in the RCMs)

These aspects are already present in the manuscript here and there but not really given full attention.

I have struggled to understand the goal of ensuring "that the RCM-modeled melt aligns with in-situ observations". Is the motivation here to suppress/separate differences in melt rates which are due to temperature biases? I see the danger that by applying such a first order bias correction, you might blur important spatial patterns (I find Fig. A1 quite informative). Another strategy might have been to diagnose the AWS temperatures for which melt typically is detected by ASCAT, (instead of the individual RCMs) and diagnose which melt rates are typically produced if this temperature is simulated- please motivate your choice. Also, are the melt thresholds a good choice for all regions, altitudes and seasons? Maybe put some additional figures in supplement to illuminate this.

Figures could be improved by introducing color scales with discrete colors

**Specific comments:**

**Abstract:**
it should be included which years are covered in this investigation
**Introduction**
l. 23: More precisely: Since 1992 → Between 1992 and 2020
l. 24: I think *Otosaka et al. (2023)* don't provide an estimate for the contribution of the SMB component. Please provide the reference (60% due to enhanced melt according to *van den Broeke et al., (2016)*?).
l. 33: "the only approach" → that is not true, maybe rather: the most comprehensive, or: RCM simulations agree best with observations (see Fettweis et al. 2020)
l.38: Fettweis et al. (2020) do not analyze future scenarios
l.42: the AWS network does not directly measure melt intensity
l.53: I understand that ASCAT can detect the onset of melt, but can it also detect the cessation or interruptions of melt? Is the decreased backscatter signal solely due to the presence of near surface water or would it remain low after a melt event due to changed chrystaline structure?
l.56: maybe: properties of the snow *pack*
l.57: delete: "*Refrezing of* " or rephrase
l.60: do you mean: can be weakened by moist subsurface layers?
l.62: is this really subsurface melt or rather meltwater in the subsurface?
Sect.: 2.1: I am missing the information which stations and how many measurements are included, do you use hourly or daily data- maybe include a table with station, location, elevation and number of temperature measurements going into this study
l. 89: I think here you want to point out that the same climate forcing may result in different melt products? Maybe elaborate and discuss whether it is possible to distinguish differences due to atmospheric differences in the RCMs and differences due to different representation of the snow pack.
Page 5: the different albedo schemes in HIRHAM5-ERAI and HIRHAM5-ERA5 will influence the melt production considerably – I recommend to acknowledge this also by some different naming to avoid misunderstandings. Maybe also include figures illustrating the relation between temperature, albedo and melt (e.g. as scatter plots)
l.109: Typo (R)ACMO
l. 111: delete once*: "On the lateral boundary"*

l. 113: which process/forcings influence snow grain size and impurity concentration?

l. 126: please provide some more information on the albedo scheme

l. 128: Is the full 2007-2024 period included here?

l.145: *"first and second"* redundant?

l. 150: please avoid jargon- what is a fully saturated signal?

l. 159: confusing statement, maybe you want to state, that, against expectation, no melt is detected near the margin even though melt is detected at higher elevation?

l.162: maybe: associated with bare ice outcropping?

Methods: generally: how do you deal with the spatial and temporal resolution of RCMs, ASCAT and AWS data?

l.172: how do you diagnose temperature bias by comparing melt (flux?) with 2m-temperature? And why would you? Why don't you simply compare simulated to observed temperature? I think there is an implicit intention here, which should be spelled out.

l. 172ff: A lot can be said here: 1) Please distinguish clearly between observing melt (surface temperature at melting point) and using a threshold *air* temperature as an indicator of melt (especially if mean temperatures are used). Specifically here but also anywhere else. Do you use daily maxima from hourly temperatures (I would recommend to do so…)? 2.) If we define a threshold in air temperature which marks the transition from no melt to melt- how much does it dependent on location, season, elevation? It could be instructive to diagnose threshold temperatures seasonally and locally in a similar fashion from simulated temperature and simulated melt.

3) and finally (-: … there is no secret connection between the modeled and the observed world- so observations can show anything independent of what the model simulates…

l. 176: here the authors are risking that readers are equating surface temperatures and air temperatures (more accurate: near-surface air temperature or 2m temperature). Also, surface temperature of a melting snow surface cannot be above 0°C.

l. 177: melt can also occur also at the surface when T2m < 0°C (low albedo, intense radiation…).

l. 182: the melting point is defined as the temperature (not the air temperature) at which snow/ice melts, please avoid to use this word in the context of air temperature.

l. 193: Isn't it: FPR=1-TN/(TN+FP)  ?

l. 206 + l. 207: in my understanding statements are contradicting here.

l. 210: This statement should be supported by some statistics rather than selected measurements

**Results:**

l. 214: correct: each drainage basin

l. 216: *"regridding biases"* should be introduced somewhere beforehand, in general iformation on how data are regridded

l. 226: Fig 4c → Fig. 4d?

l. 241: average maximum is here the multiyear mean of yearly maximal melt extent? Maybe rephrase.

l. 247: Provide the date of the melt event and also show only few days or weeks before and after the melt event in Fig 7b.

l. 262: maybe: indicate that HIRHAM-ERA5 overestimates melt

l. 262-263: It is also possible that the bias of HIRHAM-ERA5 is related to the albedo scheme

l. 285-293: HIRHAM-ERA5/I are forced by reanalysis only at the lateral boundaries of the Greenland Ice Sheet and still seem to express similar differences in the interior domain as ERA5 and ERAI. This might indicate that the observed differences between ERA5 and ERAI

originate from the farfield outside of Greenland- however Delhasse et al. () don't find corresponding differences in MAR-ERA5 and MAR-ERAI.

l. 305: "model parameter" would be understood as some internal parameter, which would change model behaviour; maybe: model parameters → simulated melt rates

l. 316: I don't understand retieved in this context

l. 320: it would help to know how many days earlier melt is detected. On could also produce melt datasets from RCM output which are smoothed by a 4-day moving mean, to test if the temporal averaging explains discrepancies.

Fig 4d: Are there regions where no melt is detected? Please mask these out similar to figure 6.

Fig 5: I think it would be more helpful to colorcode the melting threshold and to plot lines for fewer temperature thresholds which could be labeled with numbers.

Fig 6: Please also provide differences relative to the number of melt days

References (which are not also given in the manuscript):

Delhasse, A., Kittel, C., Amory, C., Hofer, S., van As, D., S. Fausto, R., and Fettweis, X.: Brief communication: Evaluation of the near-surface climate in ERA5 over the Greenland Ice Sheet, The Cryosphere, 14, 957–965, https://doi.org/10.5194/tc-14-957-2020, 2020.

---

## Author Comment (AC1)

Reply to Anonymous Referee #1 comments on

**"Bias in modeled Greenland ice sheet melt revealed by ASCAT"**

by Anna Puggaard, Nicolaj Hansen, Ruth Mottram, Thomas Nagler, Stefan Scheiblauer, Sebastian B. Simonsen, Louise S. Sørensen, Jan Wuite, and Anne M. Solgaard

**Dear Anonymous Referee #1,**

We first and foremost would like to thank you for your insightful comments on our manuscript. In the following, we try to follow and implement your suggestions to the best of our ability, and we sincerely believe that your review/comments have improved the manuscript. Below is a point-by-point response. To ease following the reply, we have your comments in black and our responses highlighted in **Blue**, and suggest changes to the manuscript in **Red.** Moreover, line numbers in our replies to comment refer to the updated MS.

**# *General comments**

1. It seems to me to be rather poorly considered that the data derived from ASCAT don't give an indication of meltwater in particular, but of all the water present on the surface and in the first layers of the snowpack. This is well presented in the introduction, but in some other parts of your text, this is a bit more confusing. Does it make sense, then, to compare this extent with melt extent in the models? And not the liquid water content in the first snow layers (see for instance Dethinne *et al*. 2023 and Picard *et al.*, 2022)?

We agree that this fact/deficiency of ASCAT melt detection should be clearly addressed throughout the manuscript. The ASCAT algorithm detects the presence of liquid (melt)water in the surface snowpack and not the process of melt. It is currently not possible to derive the quantity of meltwater with EO data alone, as stated in L39. For this reason, several simplifying assumptions and modelling are needed to derive the meltwater volume from ASCAT observations. This is, of course, a limitation in the ASCAT melt observations. In the manuscript, we show, however, that the ASCAT dataset is a useful addition to understanding biases in models better, especially as it is independent of the modelling output. The methods of Dethinne et al. (2023) and Picard et al. (2022) involve assimilating remote sensing datasets into more detailed modelling of surface melt processes to derive the meltwater volume. By including more observational data generally tends to improve the representation of present-day surface conditions (see Dethinne et al. (2023) and Langen et al. (2017)), there's still a great importance of having independent datasets to assess the models model output, in this case the ASCAT melt maps, which can help guide future improvements in model development. Together with the proposed revisions to the below general comments we propose to add to the introduction (L 43):

"Recent approaches, such as Dethinne et al. (2023) and Picard et al. (2022), have assimilated remote sensing datasets into more detailed modeling of surface melt processes to estimate

meltwater volume. Although including more observational data generally improves the representation of current surface conditions (Dethinne et al. (2023) and Langen et al. (2017)) there remains a strong need for independent observational dataset to assess model outputs."

Are we sure that the variables selected in the models are indeed representative of what the *satellite observes?*

While we acknowledge that the surface melt simulated by RCMs is not what ASCAT directly observes, there is a strong relationship between liquid water in the snowpack, to which the satellite is highly sensitive, and surface melt. This makes surface melt a meaningful and relevant model output to compare with the satellite data. Furthermore, as described in detail in section 2.3, we have applied a hierarchical decision tree using dynamic thresholds based on the conditions from the previous winter months to the ASCAT SIR product. This method allows us not only to distinguish between liquid water presence and absence but also to classify periods when liquid water starts to refreeze. In our comparisons between ASCAT melt maps and RCM melt extent, we focus only on instances where ASCAT detects a decrease in backscatter signal or when the signal is fully saturated, both indicating active surface melt. We do not include periods of refreezing in this comparison, as seen in Figure 4. This approach ensures that we are comparing relevant melt periods.
To convey this throughout the MS we suggest adding (Line xx):
"Using this method, we can distinguish between different stages of the melt water in the snow pack associated with melt water instead of only providing a binary melt extent"
We also suggest adding to line XX:
"Further, we combine label ST-2A and ST-2B into a binary active melt label"
We use the "ASCAT melt maps", which show the presence of liquid melt water on the ice sheet. In the MS we refrain from using "ASCAT observes melt" and will instead use "ASCAT observes presence of liquid water".

From another point of view, if we look at the problem the other way round, we could ask ourselves whether an additional ASCAT data processing step could be applied to translate them more specifically into meltwater.
As highlighted in the introduction, this is an active area of research but out of the scope of this MS. We believe that the current version of ASCAT water detection still is beneficial for informing on model biases. Hence, the data is mature to use as they are, despite the lack of quantifying melt volumes.

2. The comparison between RCMs is original for this type of comparison (not included in a "MIP" exercise) with remote sensed data but is also audacious and complicated. This kind of work require deep knowledge of how processes inside the model are modelized/parametrized to be able to compare it with other products. I recommend to deeply double check the way that components of meltwater production or water in the snowpack are determine in the 3 different models and be sure that everything is comparable. In that aspect, not including MAR and RACMO people is a risky move, or at least limit the study to HIRHAM alone. The idea is not to transform this study into an intercomparison of these 3 models, but to better understand it to highlight why the biases you mention are present.
The aim of this paper is to present a framework for evaluating the performance of RCMs using independent satellite observations, in this case using ASCAT melt maps to assess how well RCMs simulate the temporal variability of present-day melt extent. We understand the

reviewer's concern of the potential perception of the work as an intercomparison exercise. However, the emphasis of this study is on applying the same observations dataset to assess the model outputs. Although the internal processes related to meltwater production are undoubtedly important, however, we do not see the scope of the paper to explain the origins of the biases within the models' parametrizations but to assess how well each model captures melt patterns. While the suggestion to limit the study to HIRHAM alone is appreciated, we believe that including MAR and RACMO adds significant value by providing a broader context for evaluating model performance. This enables us to assess how well the intercomparison setup works across multiple models and see how each model captures melt patterns to highlight potential biases and strengths in the models' abilities to represent present-day melt, which could inform future model development and improvements. This clearly strengthens the argument for incorporate this approach in future MIP efforts.

To clarify this, we suggest adding to the introduction (at L. 67-71):

"By using ASCAT melt maps we aim to establish a framework for evaluating the performance of RCMs in simulating the temporal variability of present-day melt extent. As RCMs are often calibrated with respect to basin-wide surface mass balance, incorporating an independent satellite dataset like ASCAT melt maps enables a more comprehensive assessment of model performance. By including HIRHAM5, RACMO2.3p2, and MARv3.12, we assess how well each model captures surface melt patterns, focusing not on internal model parameterizations, e.g. albedo and near-surface temperature,  but on the representation of melt extent"

And also, in the conclusion (L. 386) we suggest to add:

"Our analysis demonstrates the value of using independent datasets like ASCAT to identify the spatiotemporal variability of RCM simulated melt. This approach complements traditional model validation methods and intercomparison exercises, which can inform future model development to better simulate ice sheet surface melt and possibly be incorporated in future MIP efforts."

**Specific comments**

**Data**

L119: For MAR add the same detail-level than the 2 other models (at least how albedo is retrieved).

We have added the name of the snow module, namely Crocus, and added a sentence about the internal computation of albedo.

L134-140 "MARv3.12 includes the snow model Crocus (Brun et al., 1992), that simulates a number of layers of snow, ice, or firn of variable thickness and energy- and mass-transports between each layer. The snow model also provides snow grain properties, which are used in combination with density, age, and type to simulate snow albedo (Brun et al., 1992; Fettweis et al., 2017; Antwerpen et al., 2022), MARv3.12 also have an albedo range for bare ice between 0.4 and 0.55 depending on the cleanliness of the ice (Fettweis et al., 2017). While both RACMO2.3p2 and HIRHAM5-ERA5 incorporate MODIS observations into the albedo computation, the surface albedo in MARv3.12 is only based on the internally computed broadband albedo (Brun et al., 1992)."

L136-137: Could you a bit more detail the SIR algorithm? As you need more measurements, is the reconstruction constant in time and in space? Are there any supplementary uncertainties bring with this method?

Multiple passes of ASCAT data are used to enhance image resolution and improve spatial coverage. Temporally combining these passes results in an averaged representation of the region. For Greenland, as mentioned in the manuscript, data from 4 days are combined (this is constant) to produce a single SIR (all-pass) product. These products average out any diurnal variations in sigma nought over ice (i.e. melt during day, refreezing during night). This process introduces additional uncertainty. Furthermore, potential azimuth angle dependencies are not considered in the construction of the SIR products.

We have added the following:

L152-154: "With the 4-day time interval diurnal variations in backscatter signal over ice, such as melt during day and refreezing during night, is averaged out, which introduces additional uncertainty. Additionally, the resolution enhanced ASCAT product may not capture short melt events in the spring and intense precipitation events, as these signals are averaged. Furthermore, potential azimuth angle dependencies are not considered in the construction of the SIR products."

**Methods**

- L170-171 : "We compare the RCM output of surface melt with observed 2 m temperature data", even if it could be obvious, if think this kind of sentence could be not so easy to understand at first read. I suggest switching some parts of the second paragraph of your method with the first one to be more readable.

We have rewritten the first part of the methods (L184-188): "To evaluate temperature biases in the RCMs, we compare the modeled output with observations from PROMICE GC-net AWS. Since melt is not directly measured at the AWS stations, we use 2m air temperature as a proxy for melt conditions, as near-surface air temperature is closely linked to melt processes. This approach allows us to identify and quantify temperature biases in each of the RCMs and assess how well the models simulates melt compared to in situ observations."

- Could you more detail how you construct the ROC- and PR-curves. Specifically, how do you determine if it is a true or false melt-day compared to AWS if you already calculate it for different temperature rate. If I understand well, you don't have directly melt rate observed at the AWS, but a guessing relates to the temperature measured, transform to melt through a lapse rate correction (which should be explained here). I think something is unclear for me here because of a lack of details.

To make it more clear how true positives, true negatives, false positives and false negative as constructed we have added the following:

L212-214 "Here we define the AWS as true. Thus, we define the TP when the RCMs and AWS agrees when melt is present, and FP is when no melt is observed both by the AWS and RCM. When melt is only observed at the AWS but not the RCM, it's defined as a FN and vice versa for FN."

- Please, explain here your method to choose the grid cell(s) corresponding to your AWSs. Is it the nearest neighbor, or the 4-nearest, …?

It is only the nearest grid cell with the center closest to the AWSs that is selected. To make it even more clear for the reader we have added the following sentence:

L185: "We compare the AWS to the RCM grid cell which has the closest center point to the AWS location"

- L195: You should reefer earlier to Fig. 5 to illustrate how understand the ROC- and PR-curves.

We have added the following to the sentence in L195 to refer earlier to the ROC-curves.

L215 "The ROC curve provides the total performance measure across all potential classification thresholds where a random model will produce a diagonal line, see Figure 5 as example"

- L207: You're talking about a threshold of -2°C (also in Table 1), but in your Figure 5, your curves are only from -1 to 1 °C. How did you determine this threshold?

We explored a wider range of temperature thresholds not only limited to what is shown on the plot. We have now updated the figures to include more temperature thresholds from -2 to 1 C. Based on referee #2 comments we have plotted less temperature thresholds to make the plot more concise.

- Table 1: Why only one melt threshold and one per AWS for temperature? Is it an average from all melt rate retrieved at each AWS? Your method needs to be better described for that too. Moreover, are you only using these few AWSs as presented in the Table 1? Why only these ones? In the AWS description section (2.1), you mentioned that you will use all the AWS, at least much more than presented in Table 1.

We use all PROMICE GC-net AWS stations located on the ice sheet that have data in the period of ASCAT. The included stations are shown in Figure 1. To make it clearer, we have made a correction to L186:

"To evaluate temperature biases in the RCMs, we compare the modeled output with observations from PROMICE GC-net AWS shown in Figure 1. "

Table 1 only gives example of temperatures at six selected AWS to showcase the spatial variability for temperature differences across the ice sheet. To make it even more clear from the figure text:

"Table 1 Melting thresholds for the different RCMs based on in situ PROMICE AWS observations of air temperature. The table also gives examples of the mean air temperature for August and July at six selected AWS and mean across all stations shown in Figure 1. The corresponding mean air temperature for July and August are also showcased for each RCM. Figure 3a-d illustrates the mean JJA air temperature for each RCM. "

- Still concerning observation from AWS, how do you manage the fact that most of the AWSs are in ablation area, and probably in the area where ASCAT cannot correctly detect the presence of liquid water, that you even mask outed? Or do you only consider AWSs inside the ASCAT mask? Otherwise, you decide to correct RCMs' melt with a threshold determine with comparison outside your area of melt comparison, which is not 100% valuable. Also, could please consider adding this mask in your first figure to well situated it compared to AWSs' localization as well as a more detailed comment on how you retrieved it.

The evaluation of RCMs vs AWS is done independently of ASCAT. Thus, AWS stations outside the ASCAT snowline mask are included since we want the RCMs to align with in-situ measurements across the entire ice sheet. The uneven distribution of weather stations are stated in the discussion L290-293 as a cause for a uneven representation. We further want to point that even if the stations in the ablation zone were excluded there's still an uneven distribution of weather stations on the ice sheet.

**Results**

\- L242-243: "ASCAT detects the increase in melt extent earlier compared to RCMs" Isn't it due to the detection of water by satellite and not directly melt (cf. 2nd general comment)?

It's true that ASCAT does not detect melt water directly, but rather the presence of liquid water. However, the satellite should not be able to detect firn aquifers as the penetration depth is not more than 1-2 m in dry snow conditions. Further to account for changes in the snowpack associated with melting form the previous melt cycles the algorithm applies a "recalibration of the winter signal" to account for these. This means that we ASCAT detects a decrease in backscatter we know it can only be associated with a increase in liquid water in the snowpack. We propose the following revision to line L272-275:

"At the beginning of the melt season, ASCAT detects the increase in the extent of liquid water 10-15 days earlier compared to when the RCMs simulates an increase in the melt extent. However, the decrease in extent of liquid water at the end of the melt season corresponds well with the modeled melt extent"

\- L249: can we talk about 'prediction' here as RACMO is prescribed by reanalyses at its lateral boundaries? Please rephrase with another verb.

We use simulates instead. L281: "Results show that using the in situ informed thresholds, only RACMO2p2.3 simulates melting of…"

\- Table 2: Could you add the mean number of melt day for both observation and RCMs to compare your RMSE. You should also compare the difference between your two methods (uniform or in situ informed threshold) to determine if the gain with one or the other is significant (or real statistical test, it's even better).

We have now performed Wilcoxon signed-rank test to test if there's significant difference between applying the two thresholds. We use the Wilcoxon signed-rank test since we have a paired sample, and we cannot assume a normal distributed. Thus, we cannot use the standard t-test to test if there's a significant difference between applying the two thresholds. For all RCMs the p-values shows a significant difference between the two thresholds and using the Rank-Biserial correlation we also get an indication that the magnitude of difference is significantly large. Further, we have added a new table to the MS stating the mean duration of melt period, mean number of melt days and the p-values when applying both the uniform threshold of 0.01 mmwe/day and the in situ informed melt threshold:

**Table 2.** Mean annual melt days and mean duration of the melt season for each model using two different thresholds and for ASCAT. The melt season duration is defined as starting when at least one grid point experiences melting. A Mann-Whitney U test was applied to assess whether there is no effect of using an in situ-informed threshold. Additionally, the Rank-Biserial Correlation (r-value) was computed to indicate the magnitude of the difference between thresholds.

| Threshold method | Mean melt days | | Duration of melt season | | p-value | r-value |
|---|---|---|---|---|---|---|
| | uniform | in situ | uniform | in situ | | |
| HIRHAM5-ERA5 | 25 | 21 | 225 | 204 | > 0.001 | 0.48 |
| HIRHAM5-ERAI | 16 | 16 | 205 | 201 | > 0.001 | 0.49 |
| RACMO2.3p2 | 24 | 18 | 365 | 344 | > 0.001 | 0.47 |
| MARv3.12 | 25 | 18 | 274 | 166 | > 0.001 | 0.45 |
| ASCAT | 18 | | 154 | | - | - |

Further we have added to the result section:

"Table 3 shows the mean annual number of melt days and the mean duration of the melt season for both the RCMs with the two thresholds applied and ASCAT. We define the start of the melt season when at least one grid point experiences melting. The melt extent using the in situ informed thresholds tends to align better with ASCAT observed mean number of melt days and the duration of the melt season. Furthermore, when the uniform threshold of 0.1 mm w.e. day$^{-1}$ is applied melting occurs in parts of the SW basin all year. We apply a Mann-Whitney U test to test if there is no effect of using an in situ-informed threshold. The p-value in Table 3suggests that it is very unlikely that there's no effect of using an in situ informed threshold."

**Discussion**

- L258-259: "Tab. 2 shows that by ensuring that the RCMs align with in situ measurements at specific locations." You cannot claim that RCMs align with measurement only by considering the RMSE. You need deeper statistical analyze to claim this.

We have now added an additional table, see above. We now see that applying a in situ informed threshold also aligns the mean number of melt days and the duration of melt season more closely.

- L274-276: Apply different melt threshold (based on the same in situ observation) for the different RCMs is also revealing a certain kind of bias in the model. Could you discuss that too in your Discussion?

We have re-written the start of the section to:

L308: "To get the most valid comparison between each RCM and ASCAT, we utilize in situ observations to assess biases and to determine an appropriate threshold for the melt extent in RCMs. By fitting each RCM to in situ observations we minimize the differences that is introduced due to model set-ups like resolution, parameterization etc.. Thus, we reduce..."

- L297-298: First you say that RACMO present the lowest albedo, then you also explain that MAR and HIRHAM have a lower albedo. Could you rephrase to better emphasize what are the differences and key features for each model/group of models?

Now corrected to:

L335: "RACMO2.3p2 is characterized by the highest surface albedo across the entire ice sheet, while MARv3.12 and HIRHAM5-ERAI are dominated by lower surface albedo, especially in the accumulation zone."

- L300: Could you investigate a bit more why you have such differences in RACMO and HIRHAM MODIS-based albedo in the ablation area? If it's possible, it could be nice too also compare the different albedo of the models to the MODIS albedo.

The scope of this paper is to showcase how ASCAT melt extent can be used to evaluate RCMs' performance in simulating melt extent. While albedo is an important factor influencing surface melt, our focus is on assessing melt extent directly and not on investigating albedo variability across models. Further studies could build on this work by exploring the impact of albedo differences on model performance.

Based on this comment and the 2. general comment we suggest adding a paragraph to the introduction, see 2. General comment.

- You do not talk about the differences on how the models represent the firn layer, whereas in you introduction you mention that *"The magnitude of the decrease in backscatter varies due to factors such as the snow water content and the specific properties of the snow, such as grain size and the presence of ice layers and lenses, which influence the dielectric properties and roughness geometries (Wismann, 2000; Long, 2017). "* I heard there that the signal to detect (melt)water at the surface is dependent of the snowpack conditions which are not represented/modelled in the same way in the 3 models. It should be a supplementary discussion point as melt event, and presence of water, could be delay, or more or less important, due these different way to model/parametrize the firn layers, then lead to difference when compared to ASCAT.

We agree that the model has different implementations of the surface snow/firn properties, including grain size, which could inform us about why they act differently in the evaluation. However, as the scope of the MS is not to conduct a full MIP, we here try to stay objective and see how state-of-the-are processing of ASCAT data can be utilized. It will complicate matters significantly if we model snow properties that need to be introduced in the ASCAT data, and we would not be able to separate the model biases from the observational biases. We suggest adding at L366: "Finally, the ASCAT backscatter varies due to additional factors such as specific properties of the snow, e.g. grain size and the presence of ice, due to its influence on the dielectric properties and roughness geometries. Here, two possibilities consist in progressing the melt retrievals of ASCAT; we could use the surface properties from the individual RCMs or in situ observations. For the latter, there is a seasonal bias in the in situ observations and a lack of spatial coverage, making this difficult to use for ice-sheet-wide earth observation data production. As for the RCMs implementation, this would hamper the ASCAT data as an independent data record."

- Figure 3 a-d: center your color bar to 0, it's misleading as it is. And please use only 2 varying colors, one for positive and the other for negative values. Also, please avoid yellow at pivotal value.

We have revised accordingly. For figure 3 a-d the colorbar is now centered at zero. Further, we have chosen another colorbar without yellow as a pivotal color for Figure 3, 6 and A1. All colorbars are now segmented colorbars, as suggested by referee #2. We refer to the updated MS for the updated figures.

- Figure 3, RCMs' Albedo: concerning the MAR model, you plot albedo for entire land areas and not only what looks like an ice mask in the 2 other models. Are you sure you plot the albedo used in the melt calculation, meaning the one for the ice grid points? Concerning the albedo from RACMO, considering the intercomparison and preliminary feedbacks from the PROTECT project, the albedo from RACMO presented here suggests high values, hinting a potential error when choosing which albedo plot.

We have updated all plots in figure 3 and A1 to only include data where we have ASCAT observations. Concerning the RACMO albedo, we have chosen to use the albedo provided directly by the RACMO development team. This ensures consistency with the model's intended output. Regarding the preliminary feedback from the PROTECT project, as they are not yet published, we are unable to comment or incorporate these findings into the current analysis. Once published, future studies may consider these insights for further comparison and refinement.

**Conclusions**

- L340: "[...] *can lead to more accurate simulations of surface energy balance.*" Could you rephrase, as you don't actually look at the entire surface energy balance, but only some components.

We acknowledge that we do not assess all components of the surface energy balance and how they affect the meltwater production. Instead, we focus on key variables such as radiation and albedo. Specifically, we show that by ensuring the variability of albedo is accurately simulated by the models—such as through the incorporation of MODIS—it can contribute to a more accurate representation of the surface energy balance overall. To make this more clear we have rephrased the sentence to the following:

L383: "By ensuring that the models accurately simulate the variability in albedo, such as through incorporating MODIS bare ice data, it can lead to a more accurate representation of the surface energy balance, and consequently, meltwater production."

**Appendix**

Figure A1 is exactly the same than Figure 6. Is it necessary as the appendix are in the continuity of the text and not in another document as Supplements?

This is a typo. Figure A1 is not the same as Figure 6, but rather the melt extent when a uniform threshold of 0.01 mmeq/day is applied to all models. The figure text is now updated to:

"The mean annual number of melt days modeled by the RCMs using an in uniform melt threshold of 0.01 mmeq/day to define days with significant melt. Pixels with <1 day of melt on average are marked as white, showcasing areas where melt rarely occurs. (e-h) The mean annual difference between the number of melt days in ASCAT and RCMs areas above the 2007-2020 maximum snowline elevation (Fig. 4d). Red areas correspond to more melt days in ASCAT on average and blue areas correspond to more melt days in the RCM on average. Melt in ASCAT is defined as Label ST-2A and ST-2B."

**# Technical corrections**

- L109 ACMO2.3p à RACMO2.3p2;

We have now corrected accordingly.

- L111 2x in a row "On the lateral boundary,";

We have now corrected accordingly.

- L115 2x "." in a row;

We have now corrected accordingly.

- L144-145 : 2 times in a row : "the first and second";

We have now corrected accordingly.

- Caption of Table 1: There is something wrong in this sentence: "Melting thresholds for the different RCMs based on in situ PROMICE AWS observations of 2m temperature and mean air temperature for August and July simulated by the RCMs at AWS stations and observed by the AWS stations using a lapse rate correction." I think you need to remove 'and observed by the AWS stations'.

*Agree, we revised accordingly. See above response in the section about methods.*

- L232: close the bracket here: "(Fig. 6.";

We have now corrected accordingly.

- L357: HIMHAM5 dataàHIRHAM5 data.

We have now corrected accordingly.

**References**

Dethinne, T., Glaude, Q., Picard, G., Kittel, C., Alexander, P., Orban, A., & Fettweis, X. (2023). Sensitivity of the MAR regional climate model snowpack to the parameterization of the assimilation of satellite-derived wet-snow masks on the Antarctic Peninsula. The Cryosphere, 17(10), 4267-4288.

Picard, G., Leduc-Leballeur, M., Banwell, A. F., Brucker, L., & Macelloni, G. (2022). The sensitivity of satellite microwave observations to liquid water in the Antarctic snowpack. The Cryosphere, 16(12), 5061-5083.

---

## Author Comment (AC2)

Reply to Anonymous Referee #2 comments on

**"Bias in modeled Greenland ice sheet melt revealed by ASCAT"**

by

Anna Puggaard, Nicolaj Hansen, Ruth Mottram, Thomas Nagler, Stefan Scheiblauer, Sebastian B. Simonsen, Louise S. Sørensen, Jan Wuite, and Anne M. Solgaard

**Dear Anonymous Referee #2,**

We first and foremost would like to thank you for your insightful comments on our manuscript. In the following, we try to follow and implement your suggestions to the best of our ability, and we sincerely believe that your review/comments have improved the manuscript. Below is a point-by-point response. To ease following the reply, we have your comments in black and our responses highlighted in **Blue**, and suggest changes to the manuscript in **Red.** Moreover, line numbers in our replies to comment refer to the updated MS.

**General remarks:**

1. Essential pieces of information are missing in the method part (in some cases these are implicitly given in the text): which period is taken into account, what frequency is analysed (hourly, daily, other?), how is different spatial and temporal resolution in the different data sets treated? How are datasets regridded? There are regridding biases mentioned- these could be illustrated or estimated.

We agree that this information is either missing or unclear. Thus, we have revised the beginning of the method section (L184): "To evaluate temperature biases in the RCMs, we compare the modeled output with daily observations from 2007 to 2020 by the PROMICE GC-net AWS shown in Figure 1."

To clarify the process of regridding we have added to the end of the method section (L230-234):

"To ensure consistency across datasets, all RCMs are regridded to a common grid, in this case the ASCAT grid. For HIRHAM5 and RACMO2.3p2, we apply the nearest neighbor interpolation method, while for MARv3.12 we use a cubic interpolation method. It is important to note that regridding can potentially introduce a bias within the RCM output, meaning systematic errors not associated with internal parameterization choices within the RCMs. These potential regridding biases are taken into account when comparing to the ASCAT melt extent"

2. In my view the introduction could point out more clearly the potential benefit of the ASCAT data set: Surface mass balance estimates have been mostly evaluated and also tuned with respect to the (basin wide) mass balance of the GrIS. Potential biases in melt rates above the snow line might be overlooked like that, as these are not necessarily resulting in mass changes. However these regions, where melt occurs only sporadically today, might turn into regions which contribute to sea-level rise in the near future.

This is very true so thank you for your insights. We have included the following in L72-75: "By using ASCAT melt maps we aim to establish a framework for evaluating the performance of RCMs in simulating the temporal variability of present-day melt extent. As RCMs are often calibrated with respect to basin-wide surface mass balance, incorporating an independent satellite dataset like ASCAT melt maps enables a more comprehensive assessment of model performance."

3. It seems overambitious to try to investigate why different RCMs represent melt rates differently without a much deeper dive into characteristics of snow properties and climate forcing. On the other hand, to show that the different models simulate different melt extents does not require a satellite data set.

The aim of our paper is not to explore why RCMs simulate melt rates differently. Instead, we want to show how we can use satellite-derived melt extent (e.g., ASCAT) to evaluate the performance of RCMs in simulating melt extent. This approach allows us to assess how well the models reflect the present-day melt extent, providing a critical observational benchmark. As you mentioned, RCMs are often tuned with respect to basin-wide or point-wise surface mass balance. By incorporating an independent satellite dataset, such as ASCAT, we can make a more comprehensive assessment of model performance. This provides a broader and more comprehensive assessment of how accurately the models simulate melt patterns across the ice sheet.

Based on your comment and the 3$^{rd}$ general comment of reviewer #1, we suggest an addition to both the introduction and conclusion, see reply to reviewer #1, to clarify the scope of MS

Instead I would propose to emphasize and focus on questions like:
Is the onset of melt detected too late systematically? How many days?
Are there differences between regions which experience surface melt every year and regions where melt occurs only in extreme melt years like 2012.
Is the length of melt periods overestimated by ASCAT due to the residual meltwater in the snow pack? If so, is this bias particularly pronounced for long, intense or short periods? Where melt periods occur intermittently within one year, are later melt periods represented differently than early melt periods (idea: the albedo might not recover fully after a melt event and the snow surface might be more vulnerable- which is potentially not represented in the RCMs)
These aspects are already present in the manuscript here and there but not really given full attention.

Yes, these are interesting features to discuss. Table 2 and the Results section address many of these questions on a basin scale and a discussion of the observed differences is included in Section 5.1. However, we include more details on the difference in onset of melt in Section 4 (L.275): "At the beginning of the melt season, ASCAT detects the increase in the extent of liquid water 10-15 days earlier compared to when the RCMs simulates an increase in the melt extent."

In section 5.2 we have added the following:

"However, based on the Figure 7 we asses that on average ASCAT detects the liquid water more than 4 days before the RCMs simulates melt, meaning that the processing averaging cannot fully explain the differences between ASCAT and RCMs."

And further in section 5.2:

"On average, the magnitude of the melt seasonal cycle of melt extent agrees well with RCMs, suggesting that luquid water is observed earlier but at similar extent."

We lastly we include some of these observations in the first paragraph of the Conclusions section:

"For all RCMs included in this study, the onset of melt occurs more than a week later than what is observed from ASCAT even when accounting for the averaging of satellite data. Further we see that the regions with the largest differences in total number of melt days across all RCMs are the SW and SE basins of the ice sheet."

4. I have struggled to understand the goal of ensuring "that the RCM-modeled melt aligns with in-situ observations". Is the motivation here to suppress/separate differences in melt rates which are due to temperature biases? I see the danger that by applying such a first order bias correction, you might blur important spatial patterns (I find Fig. A1 quite informative). Another strategy might have been to diagnose the AWS temperatures for which melt typically is detected by ASCAT, (instead of the individual RCMs) and diagnose which melt rates are typically produced if this temperature is simulated- please motivate your choice. Also, are the melt thresholds a good choice for all regions, altitudes and seasons? Maybe put some additional figures in supplement to illuminate this.

To derive the melt extent from RCMs we need to used a melting threshold. By using the uniform 0.1 mmwe/day we favor models that simulates low melt. Instead we perform an indepent comparison using in situ observation to infer a melting threshold. To make this more clear we have added the following to the beginning of section 5.1 (L308): "To get the most valid comparison between each RCM and ASCAT, we utilize in situ observations to assess biases and to determine an appropriate threshold for the melt extent in RCMs. By fitting each RCM to in situ observations we minimize the differences that are introduced due to model set-ups like resolution, parameterization etc. Thus, we reduce overall inter-model discrepancies as well as differences in melt extent compared to that observed by ASCAT. Despite applying the in situ informed thresholds, persistent patterns between RCMs and ASCAT remain."

5. Figures could be improved by introducing color scales with discrete colors

We have revised accordingly. All colorbars are now segmented colorbars. Further, we have chosen another colorbar without yellow as a pivotal color for Figure 3, 6 and A1, as suggested by referee #1.  We refer to the updated MS for the updated figures.

**Specific comments:**
**Abstract:**
it should be included which years are covered in this investigation

We have revised accordingly. L4-6: "Here, we explore novel processing of data from the Advanced SCATterometer (ASCAT) instrument onboard the EUMETSAT Metop satellites, which provides estimates of the spatiotemporal variability of melt extent over the Greenland Ice Sheet between 2007 and 2020"

**Introduction**

l. 23: More precisely: Since 1992→Between 1992 and 2020
We have revised accordingly.

l. 24: I think *Otosaka et al. (2023)* don't provide an estimate for the contribution of the SMB component. Please provide the reference (60% due to enhanced melt according to *van den Broeke et al., (2016)*?).
We have revised the references accordingly: L22-24 "Between 1992 and 2020, satellite observations have shown that the Greenland Ice Sheet has lost 4892 ± 457 Gt of ice or 13.6 ± 1.3 mm sea level equivalent (Otosaka et al., 2023) with half of the mass loss attributed to a decrease in the surface mass balance (SMB, van den Broeke et al., 2016)"

l. 33: "the only approach"→that is not true, maybe rather: the most comprehensive, or: RCM simulations agree best with observations (see Fettweis et al. 2020)
We have revised accordingly. L34: "At present, regional climate models (RCMs) provide the most comprehensive approach for obtaining ice-sheet-wide estimates of meltwater volumes and runoff, with simulations showing the best agreement with observations (Fettweis et al., 2020)."

l.38: Fettweis et al. (2020) do not analyze future scenarios
We have revised accordingly and found a more suiting reference. L40: "… representing differences in parametrizations, have large effects on projections of melt, runoff and surface mass balance when run into the future, giving greater uncertainty on sea level rise estimates than desirable for climate adaptation purposes (Goelzer et al., 2020)"

l.42: the AWS network does not directly measure melt intensity
We have corrected so the statement makes it clear that melt intensity has be derived from AWS stations, but not observed directly:
L42-44: "While melt intensity can be derived from in-situ observations at automatic weather stations (AWS), the sparse distribution of these stations across the ice sheet limits the evaluation of melt estimates beyond local scales (Fausto et al., 2018)."

l.53: I understand that ASCAT can detect the onset of melt, but can it also detect the cessation or interruptions of melt? Is the decreased backscatter signal solely due to the presence of near surface water or would it remain low after a melt event due to changed chrystaline structure?
This is described in full detail in the ASCAT melt maps section 2.3

l.56: maybe: properties of the snow *pack*
We have revised accordingly.

l.57: delete: "*Refrezing of* " or rephrase
We have revised accordingly (L: 63): "Refrozen meltwater from the previous melt season can percolate into the firn"

l.60: do you mean: can be weakened by moist subsurface layers?

To convey this more clear in the MS we have edited the sentence (L65): "Further, meltwater in the subsurface can still be detected after refreezing of the surface layer as the low-frequency signals can still penetrate into the refrozen surface layer"

l.62: is this really subsurface melt or rather meltwater in the subsurface?
See above for the suggested edit.

Sect.: 2.1: I am missing the information which stations and how many measurements are included, do you use hourly or daily data- maybe include a table with station, location, elevation and number of temperature measurements going into this study.
Agree. We have revised the beginning of the method sections: "To evaluate temperature biases in the RCMs, we compare the modeled output with daily observations from 2007 to 2020 by the PROMICE GC-net AWS shown in Figure 1."

l. 89: I think here you want to point out that the same climate forcing may result in different melt products? Maybe elaborate and discuss whether it is possible to distinguish differences due to atmospheric differences in the RCMs and differences due to different representation of the snow pack.
This is a good point, that even though the three RCMs are forced with the same reanalysis data the results can differ. This is due to model setups and parameterizations, some of these differences are described in the three subsections "HIRHAM5", RACMO2.3p2", and "MARv3.12" in the manuscript, some of these subsections have been updated in this round of review. Regarding the distinction between different atmospheric models and different snowpack schemes; it is not really possible to say where the differences arise from when we do not have a combination of the different RCMs with the different snowpack schemes. However, as stated earlier the aim of this study is not to explore why RCMs simulate melt rates differently. Instead, we want to show how we can use ASCAT melt extent. We have re-written the sentence (L.99): "However, different model setups such as horizontal and vertical resolutions, choices of parameters like surface albedo and subsurface schemes impact the surface energy balance simulated"

Page 5: the different albedo schemes in HIRHAM5-ERAI and HIRHAM5-ERA5 will influence the melt production considerably – I recommend to acknowledge this also by some different naming to avoid misunderstandings. Maybe also include figures illustrating the relation between temperature, albedo and melt (e.g. as scatter plots)
It's very true that the different albedo schemes will likely influence the melt production, which is also discussed in detail in the discussion section of the MS. We have chosen the naming conventions as concise as possible to maintain clarity throughout the paper.

l.109: Typo (R)ACMO
We have corrected accordingly.

l. 1l1: delete once: "On the lateral boundary"
We have corrected accordingly.

l. 113: which process/forcings influence snow grain size and impurity concentration?

Snow grain size can change due to multiple reasons like, the thaw/freezing cycle, compaction/settling. The impurities can come from dust.

l. 126: please provide some more information on the albedo scheme
Based on this comment and a similar comment from referee #1 we have added the following to the description of MARv3.12 (L.135-141): "MARv3.12 includes the snow model Crocus (Brun et al., 1992), that simulates a number of layers of snow, ice, or firn of variable thickness and energy- and mass-transports between each layer. The snow model also provides snow grain properties, which are used in combination with density, age, and type to simulate snow albedo (Brun et al., 1992; Fettweis et al., 2017; Antwerpen et al., 2022), MARv3.12 also have an albedo range for bare ice between 0.4 and 0.55 depending on the cleanliness of the ice (Fettweis et al., 2017). While both RACMO2.3p2 and HIRHAM5-ERA5 incorporate MODIS observations into the albedo computation, the surface albedo in MARv3.12 is only based on the internally computed broadband albedo (Brun et al., 1992)."

l. 128: Is the full 2007-2024 period included here?
No, while the satellite is operational since 2007 to present, the SIR data product only runs from 2007 to 2020. To make this more clear we have revised L158: "The ASCAT SIR product is available from 2007 to 2020 and is used to identify four different melt stages by applying a hierarchical decision tree using dynamic thresholds based on the previous winter reference month…"

l.145: "*first and second*" redundant?
We have corrected accordingly.

l. 150: please avoid jargon- what is a fully saturated signal?
We don't believe that a 'fully saturated signal' is jargon. Further, the meaning is explained in the same sentence: "an increase in the melt intensity does not lead to further lowering of the backscatter signal"

l. 159: confusing statement, maybe you want to state, that, against expectation, no melt is detected near the margin even though melt is detected at higher elevation?
We have revised accordingly to make it more clear in the MS (L160): "Further, against expectation refreezing or no melting is observed in pixels close to the margin when liquid water is detected at higher elevations on the ice sheet."

l.162: maybe: associated with bare ice outcropping?
We are not sure we can follow the question stated here, in the sentence we are argue for a removal (outcropping) of bare ice. Please elaborate this point.

**Methods:**
generally: how do you deal with the spatial and temporal resolution of RCMs, ASCAT and AWS data?
Se answer to 1st general comment, where we address this.

l.172: how do you diagnose temperature bias by comparing melt (flux?) with 2m-temperature? And why would you? Why don't you simply compare simulated to observed temperature? I think there is an implicit intention here, which should be spelled out.

The aim is to quantify a threshold for melt extent in the RCMs. Instead of choosing an arbitrary melting threshold we used in situ observations. However, PROMICE GC-net Aws do not observe melt directly. Therefore, we use temperature as a indicator of melt. Based on your comment and a similar comment by referee #1, we have edited the methods to: (L 185) "To evaluate temperature biases in the RCMs, we compare the modeled output with observations from PROMICE GC-net AWS. Since melt is not directly measured at the AWS stations, we use 2m air temperature as a proxy for melt conditions, as near-surface air temperature is closely linked to melt processes. This approach allows us to identify and quantify temperature biases in each of the RCMs and assess how well the models simulates melt compared to in-situ observations."

l. 172ff: A lot can be said here: 1) Please distinguish clearly between observing melt (surface temperature at melting point) and using a threshold *air* temperature as an indicator of melt (especially if mean temperatures are used). Specifically here but also anywhere else. Do you use daily maxima from hourly temperatures (I would recommend to do so...)?

See above reply to comment and what we suggest changing the beginning of the method section of the MS.

2.) If we define a threshold in air temperature which marks the transition from no melt to melt- how much does it dependent on location, season, elevation? It could be instructive to diagnose threshold temperatures seasonally and locally in a similar fashion from simulated temperature and simulated melt.

While it's definitely interesting to diagnose the temperature thresholds seasonally and locally we consider that beyond the aim of the MS. In order to derive the melt extent, we need to apply a threshold. The goal of using the in situ temperature measurements are to get an indication about what threshold to apply while we also evaluate the temperature biases in the RCMs ice sheet wide. With the suggested changes to the MS we believe this becomes more clear.

3) and finally (-: ... there is no secret connection between the modeled and the observed world- so observations can show anything independent of what the model simulates...

We agree, however, we can't find any place in the MS where this is suggested to be otherwise, could you please expand on this statement.

l. 176: here the authors are risking that readers are equating surface temperatures and air temperatures (more accurate: near-surface air temperature or 2m temperature). Also, surface temperature of a melting snow surface cannot be above 0°C.

We suggest making the following change to the MS (L194-198): "Air temperature is strongly correlated with melt since melt is a response to a positive surface energy balance, which occurs when the surface temperature is reaches than 0°C (Cuffey and Paterson, 2010). However, it's important to note that air temperature and surface temperature are not the same; while air temperature influences surface conditions, surface temperature depends on a combination of energy exchanges at the surface. Additionally, the local properties of the

snowpack can also affect when melt occurs, and melt can occur in the snowpack when air temperatures are below 0°C."

l. 177: melt can also occur also at the surface when T2m < 0°C (low albedo, intense radiation...).
See response to above comment, where the edits also takes this comment into account.

l. 182: the melting point is defined as the temperature (not the air temperature) at which snow/ice melts, please avoid to use this word in the context of air temperature.
We have corrected accordingly (L.202). "We explore various thresholds for temperature observations to account for other factors in the snowpack that influence when surface melt occurs"

l. 193: Isn't it: FPR=1-TN/(TN+FP) ?
yes, we have corrected accordingly.

l. 206 + l. 207: in my understanding statements are contradicting here.
They are not contradicting. In 206 we talk about the temperatures from the ROC-curve but in 207 we talk about the PR-curve. However, we do agree that this should be stated more clear (L.228): "For MARv3.12 and RACMO2.3p2 the ROC-curve suggest using temperature thresholds between 0.5 to 1 C to find the best RCM melting threshold, but the PR-curve the suggest a lower temperature threshold between -1.0 to -2.0 C yields better results."

l. 210: This statement should be supported by some statistics rather than selected measurements
We have put this statement in the results since we believe it fit better. Further we have revised the statement (L238): "However, the mean air temperature at selected stations suggest that the temperature bias is not systematic across the ice sheet."

**Results:**
l. 214: correct: each drainage basin
we have corrected accordingly.

l. 216: "*regridding biases*" should be introduced somewhere beforehand, in general iformation on how data are regridded
See reply to first general comment.

l. 226: Fig 4c→Fig. 4d?
we have corrected accordingly.

l. 241: average maximum is here the multiyear mean of yearly maximal melt extent? Maybe rephrase.
We have rephrased the sentence: (L.275): "The maximum melt extent is on average approximately 30 % of the ice sheet, except for HIRHAM5-ERA5 with >35 %."

l. 247: Provide the date of the melt event and also show only few days or weeks before and after the melt event in Fig 7b.

We have revised accordingly. L 281: "On 12. of July 2012, an extreme melting event was observed across almost the entire ice sheet (Nghiem et al., 2012)."

l. 262: maybe: indicate that HIRHAM-ERA5 overestimates melt

We have revised accordingly. L 295: "The melt threshold in HIRHAM5-ERA5 is considerably higher than the remaining melt estimates, indicating that HIRHAM-ERA5 overestimates melt."

l. 262-263: It is also possible that the bias of HIRHAM-ERA5 is related to the albedo scheme

Yes, it is very possible that it is the albedo, as HIRHAM5-ERAint uses albedo observation where HIRHAM5-ERA5 uses internally computed albedo.

l. 285-293: HIRHAM-ERA5/I are forced by reanalysis only at the lateral boundaries of the Greenland Ice Sheet and still seem to express similar differences in the interior domain as ERA5 and ERAI. This might indicate that the observed differences between ERA5 and ERAI originate from the farfield outside of Greenland- however Delhasse et al. () don't find corresponding differences in MAR-ERA5 and MAR-ERAI.

An important factor to consider is that regional climate models are often recalibrated to fit present-day ice sheet observations when forced with different reanalysis datasets. This recalibration process helps account for biases in the reanalysis. While Delhasse et al (2020). may not have explicitly described such recalibration in their study, it's likely that it was performed, which could explain why MAR doesn't show the same sensitivity to ERA5 and ERAI as HIRHAM5 does. The recalibration would mitigate the impact of any farfield differences in the reanalysis data on the MAR model. Another thing to remember, as written above, the two HIRHAM5 simulations, does not only have different forcings at the bounderies, but also different albedo schemes, one based on observations and the other one based on internally modeled albedo. Yes, it is very possible that it is the albedo, as HIRHAM5-ERAint uses albedo observation where HIRHAM5-ERA5 uses internally computed albedo. As the SMB model in HIRHAM5 is run offline, the albedo difference does of cause not explain the temperature difference, but it is likely to be the driving difference for the melt.

l. 305: "model parameter" would be understood as some internal parameter, which would change model behaviour; maybe: model parameters→simulated melt rates

We have revised accordingly (L344-345): "By aligning simulated melt rates more closely with observational data, we can improve the model estimates of meltwater production and ultimately runoff."

l. 316: I don't understand retieved in this context

Using the hierarchical decision tree approach allows us to classify refreezing in the ASCAT signal. However, we compare this to melt in the RCMs we do not include periods identified as refreezing by ASCAT. Based on a first general comment by referee #1 we have made several alterations to the description of ASCAT melt maps. Further to clear up any confusion we have edited the following: (L345): Since the refreezing periods identified from ASCAT data are not included in the melt season analysis…"

l. 320: it would help to know how many days earlier melt is detected. On could also produce melt datasets from RCM output which are smoothed by a 4-day moving mean, to test if the temporal averaging explains discrepancies.

We have included an estimate of the number of days ASCAT detects liquid water earlier compared to RCMs simulates melt. Since it's more than 4-days we do not compute the 4-day averages for the RCMs since this processing averaging cannot fully explain the difference. We have added the following to the result section (L276): "At the beginning of the melt season, ASCAT detects the increase in the extent of liquid water 10-15 days earlier compared to when the RCMs simulates an increase in the melt extent."

Fig 4d: Are there regions where no melt is detected? Please mask these out similar to figure 6.

We have now revised accordingly so that figure 4d have a segmented colorbar and zero melt days is shown as white. See MS for reference.

Fig 5: I think it would be more helpful to colorcode the melting threshold and to plot lines for fewer temperature thresholds which could be labeled with numbers.

We have updated the figure, so it includes fewer temperature lines and we have labeled the chosen melt threshold so that it is included in the figure now. We refer to the revised MS for the updated figure.

Fig 6: Please also provide differences relative to the number of melt days

We considered including a figure showing differences relative to the number of melt days; however, to keep the manuscript concise and focused, we limit the figure to number of melt days and differences in number of days. That way we want to keep a focus on where there is a high difference in number of melt days.

**References (which are not also given in the manuscript):**
Delhasse, A., Kittel, C., Amory, C., Hofer, S., van As, D., S. Fausto, R., and Fettweis, X.: Brief communication: Evaluation of the near-surface climate in ERA5 over the Greenland Ice Sheet, The Cryosphere, 14, 957–965, https://doi.org/10.5194/tc-14-957-2020, 2020.

---

## Author Response (AR2)

Reply to Anonymous Referee #1 comments on

**"Bias in modeled Greenland ice sheet melt revealed by ASCAT"**

by Anna Puggaard, Nicolaj Hansen, Ruth Mottram, Thomas Nagler, Stefan Scheiblauer, Sebastian B. Simonsen, Louise S. Sørensen, Jan Wuite, and Anne M. Solgaard

**Dear Editor and Anonymous Referee #1,**

We would like to thank you for your insightful comments on our manuscript. In the following, we try to follow and implement your suggestions to the best of our ability. Below is a point-by-point response. To ease following the reply, we have your comments in **black** and our responses highlighted in **Blue**, and we suggest changes to the manuscript in **Red.**

**# General comments:**

Two major comments remain not addressed properly. First, the confusion between the detection of water at the surface or in the snowpack and the actual melting process is still pervasive throughout the manuscript. ASCAT does not observe melt directly; it observes the presence of water, regardless of whether this water originates from melting, rain, or melt ponds, for instance. This is fundamentally different from the melting process itself. Throughout the manuscript, you seem to use the shorthand: liquid water at the surface = melt. However, this equivalence is overly simplistic and could confuse readers who are not fully aware of the differences or of what the sensor is actually measuring. This confusion is further reinforced by the use of terms such as "melt maps" to describe the ASCAT product (see specific comment).

We agree that there's a difference between the detection of liquid water and melt and further, that ASCAT only indirectly detects melt water. However, we apply a more sophisticated detection/classification method for liquid water than traditional methods (simple thresholding), and thus, we can distinguish between active melting (decrease and saturation of the backscatter signal) and refreezing (increase of the backscatter signal). See Figure 4 for an illustration of the backscatter signal and different classification classes. This is described in section 2.3 (L.171-173). Thus, we can more accurately than previously assume liquid water at the surface = melt. We have added a paragraph to the introduction detailing the "ASCAT melt maps" compared to other products; see the response to the second specific comment.

In order to clear up some of the confusion for the reader, we suggest changing some of the naming of the concept related to ASCAT detection and extent. We suggest making the following changes to mainly section 3.2 and the discussion:
- Melt classification -> liquid water classification in section (ASCAT melt maps)
- Melt extent -> liquid water extent in section 2.3 (ASCAT melt maps) and the discussion.
- We will also change the title of section 5.2 (L. 361) from Limitations of ASCAT melt observations to Limitations of ASCAT liquid water observations.

- Melt detection -> liquid water detection in section 5.2 (Limitations of ASCAT liquid water observations.

The use of the name "ASCAT melt maps" could be considered misleading. However, the goal of this MS is not to present the data product itself (This is done by Nagler et al., 2024) but rather to present the opportunities for using the "ASCAT melt maps" as validation of RCM melt output.  We use "ASCAT melt maps" since it's the name used in other literature (Nagler et al., 2024 & Boxall et al, 2024). We choose to retain this terminology to avoid any inconsistencies.

This brings us to the second point. I suggest that you also compare the extent of liquid water content (LWC) from the RCMs using the same method you applied to evaluate melt. The LWC should be as accessible as the melt variable in the outputs of the RCMs. Since liquid water first appears in the upper layer(s) of the snowpack before being accounted for in the melt variable, it is possible that LWC corresponds more closely to the ASCAT observations. Given that ASCAT detects the presence of liquid water, I would not be surprised if the snowpack's LWC provides a better comparison than melt, which appears later. This approach could also help explain the delays you have observed. A potential issue with LWC is that the snowpack layers are not equivalent to the three RCMs. However, since you are not examining the volume of liquid water, I believe there is still a feasible way to address this. This addition could significantly enhance your study by demonstrating whether it is more accurate to compare melt or liquid water content from an RCM when compared to ASCAT or other observational products to assess RCM performance.

We agree that it would be interesting to compare melt and liquid water content from the RCMs. However, we see several issues with the proposal.

Firstly, the LWC field is not saved consistently in time and space across all models to save disk space. Therefore, if we want to make the evaluation with the LWC, we will have to rerun several of the models to make such a comparison. Rerunning the RCMs to output LWC consistently through time and space is beyond the scope of this paper.

Secondly, as the reviewer mentioned, the RCMs do not have consistent layer schemes between models and in time. Thus, we cannot make a fair evaluation between models and against ASCAT. Moreover, the penetration depth of the ASCAT (radar) signal varies in time, but when liquid water is present on the surface, the signal cannot penetrate into the snowpack. Thus, it would not be a fair comparison between the models and ASCAT during the melt season, when ASCAT has no or little penetration into the snowpack.

Thirdly, since we are only evaluating the RCMs against ASCAT above the snowline, rain does not have a large effect compared to melt. Therefore, this cannot explain why we consistently see a delay in melt/liquid water compared to ASCAT.

Finally, the ASCAT product accounts for remaining liquid water in the snowpack from previous seasons by an annual recalibration of the winter signal pixel-by-pixel (as we write in L. 159). We propose to add the following sentence to the MS to make this statement  more clear to the reader:

L170-171: "To account for the possibility of remnant changes in the snowpack from the previous melt season, the winter signal is recalibrated annually pixel-by-pixel."

**Specific comments:**

Intro:

- L36: "However, these models are influenced by the chosen modeling approach, and substantial disparities persist among models (Rae et al., 2012; Vernon et al., 2013; Fettweis et al., 2020)." You should add now Glaude et al (2024).
  Done. The sentence now reads:
  L36: "However, these models are influenced by the chosen modeling approach, and substantial disparities persist among models (Rae et al., 2012; Vernon et al., 2013; Fettweis et al., 2020; Glaude et al., 2024)."

- If you want users of RCMs to refer to ASCAT as an independent product, it needs to be compared to other remotely sensed products that estimate the presence of meltwater. You already mention various studies and products in your introduction. A more detailed comparison of these different products with ASCAT is necessary here, even if they are not entirely independent. To use ASCAT as a reference, it is important to position it in relation to other similar products, highlighting its strengths and weaknesses.
  We agree that it's important to position the ASCAT melt maps to other products. Thus, we have added the following paragraph:
  L73-79: "Instead of using a simple threshold method, ASCAT melt maps utilize an algorithm that incorporates the temporal behavior of the backscattered signal. With this method, the ASCAT melt maps can not only detect the presence of liquid water on the surface but also distinguish between melting and subsequent refreezing of the surface meltwater. This makes ASCAT melt maps a unique product as they allow for a more fair comparison between the observed liquid melt water extent and the surface melt extent simulated by RCMs. Furthermore, by applying an annual recalibration of the winter signal, the product accounts for the formation of subsurface features from the previous melt season (Nagler et al., 2024). Again, this ensures a better classification of melt signals compared to previous products (Ashcraft and Long, 2006; Husman et al., 2023; Nagler et al., 2024)."

- As explained in the second major comment, I think than "ASCAT melt map" could be misleading. Even if you already improve the phrasing in the text by changing "ASCAT observes melt" into "ASCAT observes the presence of liquid water" , the term "melt map" is still being used. Why not use "wet-snow mask/map"? This distinction is well-established in the literature and has been discussed by some of the authors of the present manuscript.
  See the response to the first general comment.

Data:

- As you mention in the caption of the first Figure, could you also add in the text that you use AWSs included in the ASCAT melt detection domain? Also, I assume that the ASCAT melt detection domain is not the same than the mask of the maximum elevation of the snowline between 2007-2020 used to mask out part of the domain for the comparison. Could you specify whether all AWSs used to determine the melt rate threshold are within the comparison mask? I was not entirely convince by your explanation regarding the inclusion of AWS in the ASCAT mask (the maximum elevation of the snowline one). It's true that you want to evaluate RCMs' performance against observation over the entire

ice sheet, so it's justified to consider all AWSs in this comparison. But you use all the AWSs as well (including the ones outside the ASCAT mask) to determine melting thresholds, which are considered to determine the starting point of your melt event in the RCMs. I understand that it's challenging to have observations that cover the entire domain and represent each of the different area of the ice sheet adequately and equivalently, but this limitation should be explicitly discussed, especially if a significant number of AWSs are excluded. You should also verify whether including only the AWSs within the mask affects the determination of the thresholds. If it does have an impact, this should be mentioned. If it does not change the thresholds (and therefore likely does not alter the conclusions of your comparisons) you could simply state that you checked this and that it does not influence your results.

You are correct that the ASCAT melt detection domain is not the same as the snowline domain. By using the ASCAT melt detection domain, we simply remove stations that are not located on the ice sheet, i.e. stations located on peripheral glaciers and tundra. We further only included temperature observations from 2007-2020 and excluded any stations that did not have any temperature observations within this period. Thus, 34 stations were included in the study even though PROMICE-GCnet included 54 stations. In the data section of the MS we have made the following edit:

L99-100: "Here, we include the 34 stations on the Greenland Ice Sheet and have measurements of air temperature between 2007 and 2020."

The PROMICE-GCnet AWSs are not equally distributed over the accumulation and ablation zones. Thus, only looking at the AWS inside the snowline mask, we only include 34 stations, . Moreover, as we describe in L215-218 in the method section, the dataset exhibits an imbalance since there are, in general, fewer melt days than days without melt. Excluding stations below the maximum snowline extent will further amplify this data imbalance since melt occurs very infrequently at the majority of these stations. We refer to the temperature observations at KAN_U in the newly added diagram in the method section to showcase the low number of days above 0$^\circ$C. Compared to the other stations above the snowline mask, KAN_U experiences more days at t>= 0$^\circ$C due to the relatively low elevation as well as the southern location. This will, of course, affect the identification of a melt threshold. However, the difference will not only be because of a different physical signal but mostly due to a change in data distribution and imbalance.

Furthermore, as pointed out in line L198-200, this method of aligning the RCMs with AWS only ensures that the modeled melt aligns at the specific stations due to local weather conditions. Thus, we are interested in including a larger number of AWS in this alignment to maximize the location that RCMs realistically model melt – even if it is outside the domain that we compare with ASCAT. To make this point, we have made the following edit to the MS L198-202: "It is important to note that the AWS measurements are significantly affected by local scale weather conditions, so this approach only ensures that the modeled melt by the RCMs aligns primarily at these specific locations. Therefore, we include the maximum number of AWS to align with, rather than being limited to those above the snowline's maximum extent, which is where ASCAT and the inferred RCM melt extents are compared."

Method:

- Your method section is still not entirely clear to me. I suggest including a sketch or diagram to illustrate each step of your methodology, specifying what data are used at each step, and so on. This would be particularly useful if you want your method to be replicable for future studies. For example, it was unclear to me how you link the temperature at AWSs with the melt threshold. It seems that a few connections between the different parts of your method are missing. A diagram summarizing your methodology would greatly enhance clarity.

We have added this diagram (now Figure 5) at the beginning of the method section to make the evaluation prior to comparing ASCAT and RCMs more clear:

[Figure]

Furthermore, we have edited the first paragraph of the method section based on this comment and your previous comment about AWS data:

L191-204: "Modeled surface melt in RCMs is subject to large variability among models as seen in Fig. 2 and discussed in e.g Fettweis et al. (2020). The Greenland Ice Sheet SMB model intercomparison project (GrSMBMIP) suggested that discrepancies between RCMs are not systematic (Fettweis et al., 2020), thus there is a need for individual evaluation of each modeled melt volume product before we can compare the extent observed by ASCAT. We refer to Figure 5 for an overview of the evaluation taken prior to comparing RCMs with ASCAT. To establish a threshold (in mm w.e. day$-1$) to infer the melt extent from the simulated RCM melt volume, we compare it to the PROMICE GC-net AWS. With this comparison we identify how much meltwater must be in the models before we can also observe it at the AWS. We compare each AWS to the RCM grid cell, which has the closest center point to the AWS location; see Figure 1 for AWS locations. It is important to note that the AWS measurements are significantly affected by local scale weather conditions, so this approach only ensures that the modeled melt by the RCMs aligns primarily at these specific locations. Since melt is not directly measured at the AWS stations, we use 2m air temperature as a proxy for melt conditions, as near-surface air temperature is closely linked to melt processes. This approach allows us to identify and quantify temperature biases in each of the RCMs and assess how well the models simulate melt compared to in situ observations."

- Why did you use different methods to interpolate the 3 model results onto the ASCAT grid? For consistency, it would be preferable to use the same method for all three models. This choice should be justified in the text. Line 246. As you mentioned it in the text, "different method of interpolation could bring again more uncertainties" , you state that you account for this bias in your results. Could you explain more explicitly how you address the resulting bias?

For RACMO2.3p (1km) and HIRHAM5 (5km) we upscale data to get on the common grid of ASCAT (5.5km), whereas for MAR3.12 (10km) we downscale data to get on the common grid. Therefore, we use a different interpolation method. We have added this to the methods section:

L244-245: "To ensure consistency across datasets, all RCMs are regridded to a common grid, in this case, the ASCAT grid of 5.5 km resolution. For HIRHAM5 and RACMO2.3p2 we upscale data and apply the nearest neighbor interpolation method. For MARv3.12 downscale data and use a cubic interpolation method"

Concerning the introduction of systematic biases. This was mostly a concern for choosing the baseline threshold, where we want to apply a threshold to all RCMs to assess the simulated melt in the RCMs without accounting for the warm/cold biases. In theory, a melt day in the RCMs should be defined as > 0 mm w.e./day, so this baseline threshold should be the smallest possible. However, we saw that the regridding introduced a systematic positive melt volume to the model output. Thus, the aim is to choose the lowest threshold, which accounts for the possibility of small biases in the models. This ended being a threshold of > 0.1 mm w.e./day.

To make this more clear, we suggest adding:

L250-252: "The potential implication of the regridding biases is considered when choosing a baseline threshold. Here, the aim is to apply a baseline threshold to all RCMs independent of warm/cold bias within the RCMs. Therefore, the baseline threshold was set to the smallest value possible without allowing regridding biases to impact the number of melt days."

Results:
- Table 1 caption: "The corresponding mean air temperature for July and August are also showcased for each RCM. Figure 3a-d illustrates the mean JJA air temperature for each RCM. "You need to clarify in the caption which numbers are for August and for July?

We compute the mean of July and August Air temperature and not a mean for July and August separately. To clear up this confusion, we suggest changing the Table caption to the following:

Table 1: "Melting thresholds for the different RCMs based on in situ PROMICE AWS observations of air temperature. The table also gives examples of the mean July and August air temperatures at six selected AWS and the mean across all stations. The locations of all stations included in the study are shown in Figure 1. The corresponding mean July and August air temperature are also showcased for each RCM."

- L245-246. I assume the melt days are considered into a common mask. The

ASCAT one? Could you precise? And could you also precise if it is considered before or after interpolation on the common grid? By the way, these details seem more appropriate for the methods section than the results section, so I suggest moving them to the methods section to complete it.

We agree that this information is not explicitly given in the current MS and that this information belongs in the method section. Thus, we suggest adding the following to the MS in the method section:

L251: "When comparing the RCM melt extent and ASCAT liquid water extent, we apply a similar snowline mask to the RCMs as ASCAT.

Discussion
- Paragraph 2 of ASCAT limitations: If you use LWC to complete your study, you'll also be able to better explain RCMs detection delay compared to ASCAT.

  See reply to second general comments

Typs
L176: "a effect" -> an effects
Done.

L229: "the PR-curve the suggest": remove the extra "the".
Done.

Figure 3: subplot indexes (a-p) are missing on the subplots.
Done.

References:
Boxall, K., Christie, F. D. W., Willis, I. C., Wuite, J., Nagler, T., & Scheiblauer, S. (2024). Drivers of seasonal land-ice-flow variability in the Antarctic Peninsula. Journal of Geophysical Research: Earth Surface, 129, e2023JF007378. https://doi.org/10.1029/2023JF007378s

---

## Author Response (AR3)

Reply to Editor comments on

**"Bias in modeled Greenland ice sheet melt revealed by ASCAT"**

by Anna Puggaard, Nicolaj Hansen, Ruth Mottram, Thomas Nagler, Stefan Scheiblauer, Sebastian B. Simonsen, Louise S. Sørensen, Jan Wuite, and Anne M. Solgaard

Terminology of "melt" and "melt maps". I thank you for your choice to change "melt" to "liquid water" throughout, following the reviewer's insight. However, I still find use of "melt maps" to be incorrect, and now inconsistent within the new terminology of the manuscript. I agree with the reviewer that "melt map" is a misleading name, and I disagree with the authors' assertion that the terminology should be repeated in order to agree with recent work (Nagler et al., 2024 and Boxall et al, 2024). Those studies did not have the benefit of this astute reviewer comment, but this does not mean we need to propagate their flawed terminology forward. Please rename all "melt maps" to something consistent with what they are: "liquid water maps" or however you judge best to craft it.

We have now changed the name to ASCAT liquid water maps throughout the whole MS. To mitigate possible confusion about the different names, we have added this the first time ASCAT liquid water maps are introduced:

L 73-73: "Instead of using a simple threshold method, ASCAT liquid water maps, also known as ASCAT melt maps, utilize an algorithm that incorporates the temporal behavior of the backscattered signal"

**Other minor suggestions:**

"Therefore, the baseline threshold was set to the smallest value possible " -- Please state the value (0.1 mm w.e./day). These sentences also appear in both the Methods and the Results section. Please remove them from Results.

We have removed the sentences from the results and stated the value in the method section:

L249-251: "Therefore, the baseline threshold was set to the smallest value possible (0.1 mm w.e. day−1) without allowing regridding biases to impact the number of melt days."

Table 1, "the mean July and August air temperatures" -- this would be clearer as "the mean July through August air temperatures".

Done.

---

## Author Response (AR4)

Reply to Editor comments on

**"Bias in modeled Greenland ice sheet melt revealed by ASCAT"**

by Anna Puggaard, Nicolaj Hansen, Ruth Mottram, Thomas Nagler, Stefan Scheiblauer, Sebastian B. Simonsen, Louise S. Sørensen, Jan Wuite, and Anne M. Solgaard

Thank you for your feedback. We will review and update the figures as needed to improve colorblind accessibility, as indicated by the Copernicus system. Based on the **Coblis – Color Blindness Simulator**, we found that it might have been **Figure 5** that was flagged by the Copernicus system. We have updated the figure accordingly.

Regarding terminology, our co-author **Thomas Nagler**, who is responsible for generating the maps, has emphasized that **"surface melt extent"** has been the standard term in remote sensing, glaciology, and polar climate research for over 40 years. This terminology is widely used in the literature and is formally recognized in key reference
reports such as the **IGOS 2007 Cryosphere report:**
**https://www.wgms.ch/downloads/IGOS_2007.pdf, page 99**
and **NSIDC resources**:
https://nsidc.org/ice-sheets-today/melt-data-tools.
Furthermore, this point is also **stated in the current manuscript** (L57-60):
"This sensitivity to meltwater has enabled several studies to estimate melt over both ice sheets using passive and active microwave measurement with a threshold method to detect the onset of melt and its extent (Long and Drinkwater, 1994; Wismann, 2000; Ashcraft and Long, 2006; Fettweis et al., 2011; Colosio et al., 2021; Husman et al., 2023)."

Our **surface melt extent maps** detect the presence of **liquid water within the snow and ice surface layer**, where the medium consists of ice particles with a small fraction of liquid water from melt processes. Some liquid water may persist in deeper layers even after surface refreezing. However, the product does **not** detect the extent of surface lakes or drainage channels, which have distinct microwave signatures. For this reason, referring to it as **"liquid water extent"** is also misleading, as it could be confused with open water features. Therefore, we prefer to keep the legacy naming convention of the product ASCAT surface melt extent maps. **In the updated manuscript, ASCAT surface melt extent maps now appear 9 times: L6, L72, L74, L75, L81, L83, L151, L417, and Figure 5 caption.**

The previous change from **"melt maps"** to **"liquid water maps"** was not the best compromise, as it introduced ambiguity. We are very sorry, but we did not think of the possible confusion with open water features in the first round of revisions to accommodate your suggestions. Given the long-standing and well-accepted use of "surface melt extent" in the scientific community, we hope to maintain this terminology.
To directly address your comment, we will revise the sentence as follows:

**"Again, this ensures a better classification of melt signal compared to previous melt extent products from both active and passive microwave measurements over the Greenland Ice Sheet, such as Abdalati and Steffen (1995); Wismann (2000); Nghiem et al. (2001); Tedesco (2007); Fettweis et al. (2011); Colosio et al. (2021)."**

This ensures clarity and consistency with established terminology while maintaining the intended comparison to previous studies.

All references are included in the updated MS.

**On behalf of all co-authors, we are sorry for our oversight and too fast compromise of the implementation of the thermology: "liquid water maps".**

**Anna Puggaard**

---

## Author Response (AR5)

Reply to Editor comments on

**"Bias in modeled Greenland ice sheet melt revealed by ASCAT"**

by Anna Puggaard, Nicolaj Hansen, Ruth Mottram, Thomas Nagler, Stefan Scheiblauer, Sebastian B. Simonsen, Louise S. Sørensen, Jan Wuite, and Anne M. Solgaard

Thank you for your feedback. We will rename the ASCAT surface melt extent maps to ASCAT wet snow maps. Here is a list of places where we have changed the name from ASCAT surface melt extent maps to ASCAT wet snow maps:
Line 6, 73, 74, 75, 82, 84, 151, 390, and Figure 5 caption.

Based on your feedback, we've taken a closer look at how Banwell et al. (2023) approached this terminology. We assume that when you write 'melt mask', you refer to the melt maps since 'melt mask' does not appear in the current MS. Anyhow, we've decided to follow the same practice they used, as it strikes a good compromise between clarity and consistency with the existing literature. Therefore, we make the following edit to the paper:
Line 183 – "When comparing ASCAT wet snow maps and the melt extent model by the RCMs, we excluded label ST-3 (increase in the refrozen layer) in order to make the most fair comparison with RCM surface melt. Throughout this paper, we use the phrase melt days when referring to days labeled ST-2A (surface melt) and ST-2B (wet snow layer) in the ASCAT wet snow maps. Similarly, melt extent will refer to pixels labeled ST-2A and ST-2 B. It should be noted that active surface melting may not necessarily be occurring on the ASCAT-derived melt days, but meltwater may simply be present without ongoing melt. However, we mitigate the potential differences by excluding days classified as refreezing and recalibrating the winter baseline to account for liquid water that persisted through the winter."

With this added, we have also made several smaller changes to the MS, mainly in the results and discussion sections, since we can now refer to the ASCAT melt days and melt extent. We refer to the track changes document, which shows the small changes clearly.

Lastly based on your suggestion, we have changed the colormap of Figure 6, which will hopefully solve the problem: